# HiPhO: How Far Are (M)LLMs from Humans in the Latest High School Physics Olympiad Benchmark?

Fangchen Yu [1*]  Haiyuan Wan [12*]  Qianjia Cheng [13*]  Yuchen Zhang [14]  Jiacheng Chen [15]
Fujun Han [6]  Yulun Wu [13]  Junchi Yao [17]  Ruilizhen Hu [6]  Ning Ding [12]  Yu Cheng [15]  Tao Chen [8]  Lei Bai [1]
Dongzhan Zhou [1†]  Yun Luo [1†]  Ganqu Cui [1†]  Peng Ye [15†]

## Abstract

Recently, the physics reasoning capabilities of (M)LLMs have attracted growing attention. However, existing physics benchmarks lack systematic coverage of recent physics Olympiads and direct comparison with human contestants. We present **HiPhO**, the first benchmark dedicated to high school physics Olympiads with human-aligned evaluation. HiPhO highlights three key innovations. **(1) Comprehensive data:** it compiles 13 latest Olympiads from 2024–2025, covering international and regional competitions and spanning mixed modalities from text-only to diagram-based problems. **(2) Professional evaluation:** it adopts official rubrics for fine-grained answer- and step-level grading aligned with human examiners. **(3) Human-level comparison:** it assigns gold, silver, and bronze medals to models based on official medal scores, enabling direct comparison with human contestants. Evaluating 30 (M)LLMs across 13 exams, we find that most open-source MLLMs remain at or below the bronze level, open-source LLMs demonstrate notable progress with multiple gold medals, and closed-source MLLMs achieve 6-13 gold medals, while most models still fall well short of full marks. These results underscore the substantial gap between current (M)LLMs and top human contestants, as well as the room for further improvement. The dataset and leaderboard are available at https://huggingface.co/datasets/SciYu/HiPhO and https://phyarena.github.io, respectively.

*Equal contribution  [1]Shanghai AI Laboratory  [2]Tsinghua University  [3]Zhejiang University  [4]Peking University  [5]CUHK  [6]CUHK-Shenzhen  [7]MBZUAI  [8]Fudan University.  Correspondence to:  Peng Ye <yepeng@pjlab.org.cn>, Ganqu Cui <thu.cgq@foxmail.com>, Yun Luo <luoyun1@pjlab.org.cn>, Dongzhan Zhou <zhoudongzhan@pjlab.org.cn>.

*Proceedings of the 43rd International Conference on Machine Learning*, Seoul, South Korea. PMLR 306, 2026. Copyright 2026 by the author(s).

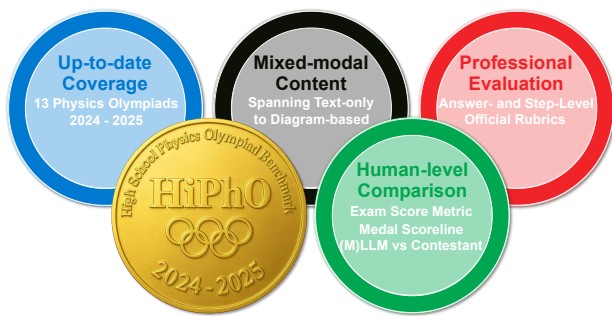

*Figure 1.* Overview of the **HiPhO** (High School Physics Olympiad) benchmark, featuring up-to-date coverage, mixed-modal content, professional step-level evaluation, and human-level comparative scoring for rigorous assessment of multimodal physics reasoning.

## 1. Introduction

Large language models (LLMs) and multimodal LLMs (MLLMs) have recently attracted growing attention for their physics reasoning capabilities. However, existing physics datasets remain limited in scope, lacking both systematic coverage of physics Olympiads and direct comparison with human contestants. In contrast, mathematical Olympiads have been extensively studied. Recent models such as GPT-5.2 achieved 100% accuracy on AIME-2025 (MAA, 2025), and Gemini Deep Think has reached gold-medal level at the 2025 International Mathematical Olympiad (IMO). As the crown jewel of physics exams, physics Olympiads provide a uniquely rigorous testbed that remains underexplored, highlighting the need for a new benchmark with systematic evaluation and human-aligned comparison.

Unlike mathematical Olympiads, physics Olympiads demand understanding of physical principles, derivation of abstract formulas, and reasoning over complex multimodal diagrams. These properties make them a strong testbed for assessing whether (M)LLMs can perform genuine visual and physical reasoning, while also enabling direct comparison with human contestants.

However, current Olympiad-related physics datasets exhibit several critical limitations: **(1) Outdated coverage:** Datasets like OlympiadBench (He et al., 2024) includes

*Table 1.* Comparison of physics benchmarks across Olympiad coverage, modality, evaluation method, metric, and use of medal scores. HiPhO introduces four key improvements: (1) up-to-date coverage of 13 Olympiads from 2024–2025; (2) fine-grained modality categorization, where 1 = text-only, 2 = text + illustration figure, 3 = text + variable figure, and 4 = text + data figure (see Fig. 3); (3) step-level grading using official rubrics; and (4) the first comparison of model performance with medal cutoffs and contestant scores.

| Benchmark | Real-World Olympiad | Multimodal Problem | Modality Type | Evaluation Method | Performance Metric | Medal Score |
|---|---|---|---|---|---|---|
| *Olympiad-related Physics Benchmarks or Subsets* | | | | | | |
| PHYBench (Qiu et al., 2026) | ✗ | ✗ | 1 | Answer | Acc., EED Score | ✗ |
| PHYSICS (Zheng et al., 2026) | ✗ | ✗ | 1 | Answer | Acc. | ✗ |
| OlympiadBench (He et al., 2024) | ✓ | ✓ | 1,2 | Answer | Acc. | ✗ |
| OlympicArena (Huang et al., 2024) | ✓ | ✓ | 1,2 | Answer+Model Step[1] | Acc. | ✗ |
| PhysReason (Zhang et al., 2025) | ✓ | ✓ | 1,2 | Answer+Error Step[2] | Acc. | ✗ |
| SeePhys (Xiang et al., 2026) | ✓ | ✓ | 1,2,4 | Answer | Acc. | ✗ |
| *Olympiad-focused Physics Benchmark* | | | | | | |
| **HiPhO (Ours)** | ✓ | ✓ | 1,2,3,4 | Answer+Rubric Step[3] | Exam Score | ✓ |

[1]GPT4-generated steps; [2]First-error-step detection in model output; [3]Official rubrics with scoring per predefined point.

International Physics Olympiad (IPhO) problems only up to 2021, omitting the most recent two years of exams. **(2) Lack of multimodal content:** Physics Olympiad problems often involve complex diagrams. However, datasets such as PHYBench (Qiu et al., 2026) and PHYSICS (Zheng et al., 2026) are entirely text-based. **(3) Limited evaluation quality:** Most datasets stop at coarse answer-level evaluation, without step-level grading aligned with official rubrics. **(4) No human-level comparison:** Existing datasets typically report accuracy, without exam-score-based comparisons against human contestants in real-world competitions.

To address these limitations, we introduce *HiPhO*, the first benchmark dedicated to recent physics Olympiads with human-aligned evaluation. As shown in Fig. 1, HiPhO incorporates four key improvements: **(1) Up-to-date coverage:** it compiles the latest 13 Olympiads from 2024–2025, including both international and regional contests. **(2) Mixed-modal content:** problems are collected as complete exams, spanning text-only to diagram-based, with fine-grained categorization of diagram types (see Fig. 3). **(3) Professional evaluation:** it adopts official rubrics for fine-grained grading at both the answer and step levels, aligned with human examiners to ensure rigorous assessment. **(4) Human-level comparison:** it computes full exam scores for models and maps them to gold, silver, and bronze medal thresholds, thereby enabling direct comparison with human contestants.

Evaluating 30 (M)LLMs across 13 exams, we observe a clear performance hierarchy. Closed-source MLLMs lead the medal table, obtaining 6-13 gold medals across 13 Olympiads. Nevertheless, even the strongest models, such as Gemini-3-Pro and GPT-5.2, remain below the very best human contestants. By contrast, leading open-source MLLMs can reach up to 10 gold medals, although most remain at or below the bronze threshold. More encouragingly, several open-source LLMs, including P1-235B-A22B and GPT-OSS-120B (high), each attain 12 gold medals,

even on hard contests such as IPhO-2025. Overall, these results highlight the strong yet non-parity performance of closed-source models, the remaining limitations of open-source MLLMs, and the promising progress of open-source reasoning (M)LLMs in advancing physics problem solving.

In summary, our main contributions are as follows:

● **Olympiad-focused Benchmark.** We present HiPhO, the first benchmark dedicated to high school physics Olympiads, comprising 13 Olympiad exams from 2024–2025. All problems are expert-curated, structurally extracted, and manually verified to ensure high quality and consistency.

● **Human-aligned Scoring.** We evaluate performance using exam scores rather than the commonly used accuracy, applying both answer- and step-level grading based on official rubrics. This enables direct, medal-standard comparisons with actual human contestants.

● **Human-level Comparison.** Compared to human contestants, closed-source reasoning MLLMs reach gold in 6-13 exams, while open-source MLLMs stay mostly at bronze; some open-source LLMs show stronger reasoning with multiple golds, yet all remain far from the very top students.

## 2. Related Work

Among Olympiad-related physics datasets (Table 1), PHYBench (Qiu et al., 2026) and PHYSICS (Zheng et al., 2026) are text-only, while OlympiadBench (He et al., 2024) and OlympicArena (Huang et al., 2024) cover outdated exams. Multimodal datasets such as SeePhys (Xiang et al., 2026) and PhysReason (Zhang et al., 2025) remain restricted in scope (e.g., IPhO or CPhO). By contrast, HiPhO compiles 13 recent Olympiads (2024-2025) with mixed-modal problems. More importantly, HiPhO goes beyond coarse answer-level accuracy by using official rubrics for fine-grained grading and exam-score metrics, enabling direct comparison between (M)LLMs and human contestants.

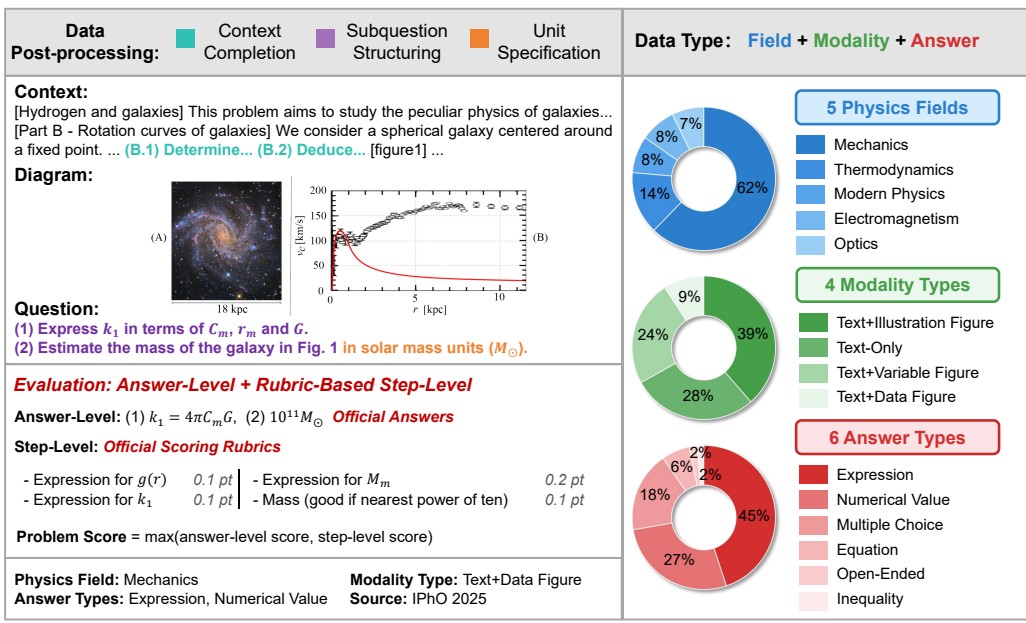

*Figure 2.* Framework and statistics of the HiPhO benchmark. **Top left:** An example problem with three post-processing steps, including context completion, subquestion structuring, and unit specification. **Bottom left:** Answer + step-level evaluation based on official rubrics. **Right:** Dataset composition across five physics fields, four modality types, and six answer types.

## 3. The HiPhO Benchmark

### 3.1. Dataset Overview

We introduce HiPhO, the first high school physics Olympiad benchmark designed to compare the reasoning capabilities of (M)LLMs against human contestants. It contributes along four key dimensions: (1) a comprehensive, up-to-date dataset; (2) mixed-modal problems; (3) fine-grained evaluation; and (4) human-level comparison.

*Table 2.* HiPhO Benchmark Statistics.

| Statistic | Count |
|---|---|
| Total Problems | 360 |
| Total Physics Fields | 5 |
| Total Modality Types | 4 |
| Total Answer Types | 6 |
| Total Difficulty Levels | 3 |
| Language (EN : ZH) | 308:52 |

**Dataset Perspective.** As shown in Table 2, HiPhO covers 360 problems from 13 Olympiad exams in 2024–2025, making it the most up-to-date benchmark. Each problem is categorized along two axes: **(1) physics taxonomy:** five fields (Mechanics, Electromagnetism, Thermodynamics, Optics, Modern Physics); and **(2) modality type:** four formats (Text-Only, Text+Illustration Figure, Text+Variable Figure, Text+Data Figure; see Fig. 3), enabling detailed analysis of physical and visual reasoning.

**Evaluation Perspective.** HiPhO introduces the first evaluation method that integrates answer-level correctness with step-level assessment using *official rubrics*. In addition, we report *exam scores* rather than accuracy to more faithfully reflect overall performance. This allows direct comparison with human scores and medal thresholds, providing the first quantitative analysis of the gap between state-of-the-art (M)LLMs and top contestants in real-world competitions.

### 3.2. Dataset Construction

**Up-to-date Coverage.** HiPhO includes 13 exams from seven major Physics Olympiads (PhOs) and competitions in 2024–2025, with details in Appendix B.1. Selection was based on influence and the availability of human scores[1]. With data processing tools, the benchmark can be efficiently updated each year, ensuring continuously refreshed coverage. We further group them into three difficulty levels: **(1) Hard:** IPhO, APhO, EuPhO; **(2) Medium:** NBPhO, PanPhO; **(3) Easy:** PanMechanics, F=MA.

**High-Quality Extraction Pipeline.** We process PDFs from official websites through a structured pipeline: **(1) Data extraction:** PDFs are converted to markdown via OCR. **(2) QA matching:** question indices align each problem with its answer. **(3) Human verification:** each QA pair is manually verified by experts to correct errors, ensuring fidelity to the original exams. **(4) Rubric structuring:** where available, step-level criteria are extracted into model-readable rubrics for fine-grained evaluation. **(5) Post-processing:** verified QAs are refined for context, question, and unit clarity.

---

[1]APhO-2024 was excluded due to the absence of human scores.

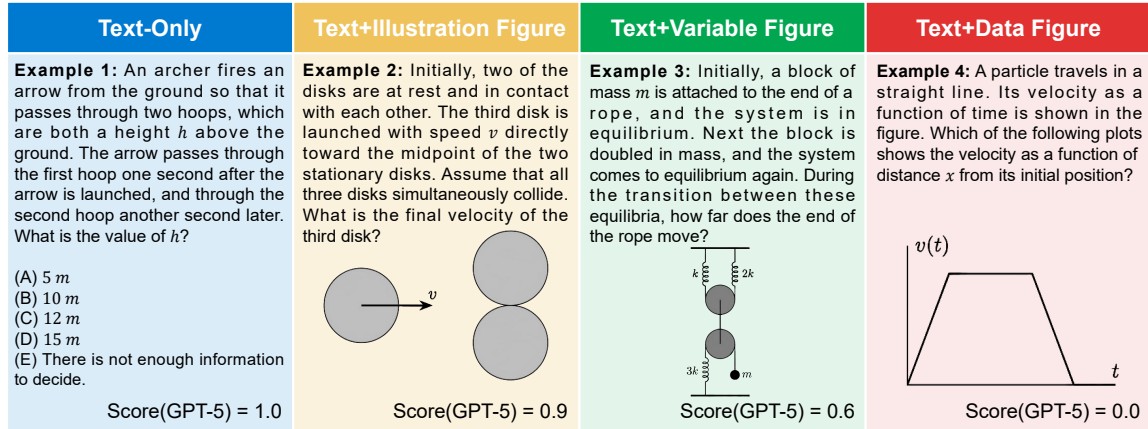

*Figure 3.* Examples of the four modality types in the HiPhO benchmark: Text-Only (TO), Text+Illustration Figure (TI), Text+Variable Figure (TV), and Text+Data Figure (TD). TO and TI rely mainly on text, whereas TV and TD require understanding variables or data, making them more challenging for visual reasoning, as reflected in GPT-5's average scores (each example worth 1.0 point).

**Post-Processing and QA Refinement.** To better reflect exam conditions and improve evaluation accuracy, we apply three steps (see Fig. 2). **(1) Context completion:** long stems with interdependent subquestions are merged for coherence. **(2) Subquestion structuring:** semantically linked subquestions are preserved and explicitly labeled to encourage ordered responses and reduce omissions. **(3) Unit specification:** required units are made explicit to avoid trivial mismatches while preserving problem difficulty.

**Comprehensive Data Annotation.** For fine-grained analysis, we annotate data along three dimensions. **(1) Physics field:** problems are labeled into five fields: Mechanics, Electromagnetism, Thermodynamics, Optics, and Modern Physics. **(2) Visual modality:** problems are grouped into four types reflecting increasing visual reasoning demands (see Fig. 3): *i. Text-Only (TO)*, purely textual statements; *ii. Text+Illustration Figure (TI)*, figures illustrate scenarios in text; *iii. Text+Variable Figure (TV)*, figures specify key variables or spatial relations; and *iv. Text+Data Figure (TD)*, figures present data, plots, or functions absent from the text. **(3) Answer type:** subquestions are classified into six forms: Expression, Numerical Value, Multiple Choice, Equation, Inequality, and Open-Ended. As shown in Fig. 2, *Expression* (45%) and *Numerical Value* (27%) dominate, highlighting symbolic reasoning and precise computation.

### 3.3. Evaluation Method

Unlike most datasets that score only final answers, our framework integrates *answer-level* and *step-level* evaluation to better reflect human grading. At the answer level, a math-verifier (Kydlíček, 2025) first checks consistency with the ground truth; answers marked inconsistent are further adjudicated by Gemini-2.5-Flash to account for equivalent alternative expressions. At the step level, model solutions are compared with official rubrics, and partial credit is awarded

for correct intermediate steps. For each problem, the final score is the maximum of its answer-level and step-level scores, preserving full credit for correct answers while fairly recognizing valid reasoning. As shown in Appendix E, step-level scoring frequently surpasses answer-level scoring, underscoring its importance in capturing intermediate reasoning and preventing systematic underestimation. We therefore report exam scores by summing over all problems, using Gemini-2.5-Flash as the default grader. Implementation details (e.g., prompts) are provided in Appendix C.

### 3.4. Data Leakage Detection

To assess potential memorization, we perform data leakage detection using the **partial-prompt completion** strategy (Wu et al., 2026). For each problem, we provide the first 60% of its content and ask the model to greedily complete the remainder with two evaluation metrics: **ROUGE-L** (Partial-Prompt Completion Rate) (Lin, 2004), measuring longest common subsequence overlap, and **ACC** (Partial-Prompt Answer Accuracy), measuring whether the continuation contains the correct answer. As summarized in Appendix B.3, ROUGE-L across 8 representative models is consistently below 15%, with average answer accuracy under 4%. Per-Olympiad results further show that across all 13 Olympiads, most completion rates remain below 20% and most answer accuracies are 0%, confirming that HiPhO exhibits minimal data leakage and remains highly challenging.

## 4. Experiments

To compare state-of-the-art (M)LLMs with top contestants, we evaluate 30 representative models, including 8 closed-source MLLMs, 11 open-source MLLMs, and 11 open-source LLMs. Experimental details are provided in Appendix D.1, with medal thresholds in Appendix D.2.

*Table 3.* Evaluation results on the HiPhO benchmark (13 physics Olympiads from 2024–2025) using the *exam score* metric. Gold, Silver and Bronze indicate scores above the respective thresholds. Models are ranked by medal counts; **bold** is the highest score, and underline is the second highest. Here, only the theoretical parts of exams are used, hence Full Mark (Model) ≤ Full Mark (Human).

| Physics Olympiad | IPhO | | APhO | EuPhO | | NBPhO | | PanPhO | | PanMechanics | | F=MA | | Medal | | |
| Year | 2025 | 2024 | 2025 | 2025 | 2024 | 2025 | 2024 | 2025 | 2024 | 2025 | 2024 | 2025 | 2024 | Table | | |
|---|---|---|---|---|---|---|---|---|---|---|---|---|---|---|---|---|
| Full Mark (Human) | 30.0 | 30.0 | 30.0 | 30.0 | 30.0 | 72.0 | 72.0 | 100.0 | 100.0 | 100.0 | 100.0 | 25.0 | 25.0 | | | |
| Full Mark (Model) | 29.4 | 29.3 | 30.0 | 29.0 | 28.0 | 43.5 | 50.0 | 100.0 | 98.0 | 100.0 | 100.0 | 25.0 | 25.0 | | | |
| Top-1 Score (Human) | 29.2 | 29.4 | 30.0 | 27.0 | 30.0 | 53.2 | 40.8 | 81.0 | 66.5 | 62.0 | 51.0 | 25.0 | 24.0 | | | |
| Top-1 Score (Model) | 25.2 | 25.9 | 28.4 | 21.0 | 23.9 | 37.1 | 43.8 | 66.3 | 82.5 | 86.0 | 82.3 | 23.8 | 23.9 | | | |
| Gold Medal | 19.7 | 20.8 | 23.3 | 16.5 | 20.4 | 28.6 | 26.5 | 41.5 | 52.0 | 52.0 | 51.0 | 15.0 | 14.0 | 🜨 | | |
| Silver Medal | 12.1 | 11.1 | 18.7 | 9.8 | 14.2 | 20.1 | 19.4 | 28.5 | 37.5 | 36.0 | 26.0 | 11.0 | 12.0 | | 🜨 | |
| Bronze Medal | 7.2 | 3.6 | 13.1 | 5.8 | 8.9 | 15.2 | 13.5 | 14.5 | 16.0 | 20.0 | 12.0 | 9.0 | 10.0 | | | 🜨 |
| *Closed-Source MLLMs* | | | | | | | | | | | | | | | | |
| Gemini-3-Pro | **25.2** | 25.2 | 28.3 | **21.0** | **23.9** | **37.1** | **43.8** | **66.3** | 76.2 | **86.0** | 74.8 | **23.8** | **23.9** | 13 | 0 | 0 |
| GPT-5.2 | 22.3 | 25.5 | **28.4** | 15.7 | **23.9** | 35.8 | 37.1 | 59.8 | **82.5** | 67.6 | **82.3** | 23.4 | 23.5 | 12 | 1 | 0 |
| Gemini-2.5-Pro | 22.7 | **25.9** | 27.9 | 14.9 | 21.8 | 32.3 | 35.9 | 60.3 | 64.1 | 69.5 | 70.2 | 22.8 | 22.0 | 12 | 1 | 0 |
| GPT-5 | 22.3 | 20.2 | 27.0 | 10.3 | 21.7 | 32.9 | 32.8 | 55.9 | 69.8 | 69.4 | 79.0 | 22.4 | 22.4 | 11 | 2 | 0 |
| o3 | 15.7 | 23.7 | 25.9 | 11.4 | 21.6 | 34.1 | 33.5 | 47.3 | 55.9 | 71.4 | 75.6 | 22.0 | 20.6 | 11 | 2 | 0 |
| Grok-4 | 18.7 | 23.5 | 25.0 | 11.5 | 20.5 | 25.8 | 29.3 | 45.0 | 75.4 | 72.1 | 78.6 | 19.8 | 19.8 | 10 | 3 | 0 |
| Claude-4-Sonnet-Thinking | 19.0 | 22.0 | 24.8 | 9.7 | 20.5 | 28.1 | 25.6 | 43.1 | 39.3 | 57.4 | 61.8 | 19.2 | 20.1 | 8 | 4 | 1 |
| o4-mini (high) | 16.0 | 23.7 | 22.9 | 12.0 | 20.1 | 27.4 | 29.8 | 41.4 | 50.9 | 69.1 | 67.3 | 18.6 | 18.8 | 6 | 7 | 0 |
| *Open-Source MLLMs* | | | | | | | | | | | | | | | | |
| Qwen3-VL-235B-A22B-Thinking | 19.8 | 23.4 | 25.4 | 10.8 | 19.0 | 29.2 | 25.9 | 48.3 | 54.0 | 64.3 | 79.9 | 21.4 | 19.9 | 10 | 3 | 0 |
| Qwen3-VL-30B-A3B-Thinking | 15.7 | 19.3 | 25.3 | 11.7 | 20.2 | 29.7 | 28.3 | 41.6 | 45.9 | 57.6 | 51.4 | 18.9 | 20.7 | 8 | 5 | 0 |
| Intern-S1 | 15.9 | 14.2 | 21.7 | 9.0 | 16.6 | 23.0 | 20.5 | 41.1 | 50.3 | 60.4 | 57.4 | 18.4 | 19.5 | 4 | 8 | 1 |
| InternVL3-78B-Instruct | 12.9 | 12.5 | 17.7 | 7.5 | 15.2 | 22.3 | 22.5 | 26.2 | 27.4 | 21.1 | 27.1 | 12.0 | 13.0 | 0 | 8 | 5 |
| GLM-4.5V | 11.9 | 4.4 | 16.2 | 8.7 | 14.1 | 19.5 | 14.0 | 18.5 | 16.0 | 47.8 | 39.0 | 13.0 | 13.8 | 0 | 4 | 9 |
| Qwen2.5-VL-72B-Instruct | 10.6 | 7.2 | 13.6 | 6.1 | 8.1 | 13.3 | 11.4 | 26.8 | 18.2 | 24.0 | 28.5 | 13.5 | 9.8 | 0 | 2 | 7 |
| LLaMA4-Scout-17B | 9.7 | 9.5 | 13.1 | 5.4 | 10.4 | 22.8 | 12.8 | 26.6 | 24.1 | 35.4 | 34.5 | 8.1 | 6.4 | 0 | 2 | 7 |
| Qwen2.5-VL-32B-Instruct | 9.9 | 8.2 | 16.5 | 6.9 | 10.0 | 15.3 | 14.4 | 22.5 | 22.4 | 28.1 | 29.9 | 7.6 | 4.6 | 0 | 1 | 10 |
| InternVL3-38B-Instruct | 8.9 | 7.8 | 12.3 | 6.1 | 8.3 | 14.0 | 10.6 | 24.1 | 20.4 | 27.5 | 24.8 | 8.2 | 6.8 | 0 | 0 | 7 |
| InternVL3-9B-Instruct | 4.7 | 3.7 | 7.2 | 4.2 | 4.1 | 9.4 | 6.0 | 12.4 | 11.0 | 11.6 | 16.3 | 6.4 | 6.2 | 0 | 0 | 2 |
| Qwen2.5-VL-7B-Instruct | 3.5 | 2.5 | 5.7 | 4.4 | 3.6 | 7.3 | 5.5 | 14.7 | 7.6 | 9.8 | 11.5 | 4.4 | 3.5 | 0 | 0 | 1 |
| *Open-Source LLMs* | | | | | | | | | | | | | | | | |
| P1-235B-A22B | 21.2 | 24.7 | 27.4 | 10.8 | 23.0 | 31.8 | 28.4 | 54.7 | 56.7 | 74.7 | 72.9 | 20.9 | 19.4 | 12 | 1 | 0 |
| GPT-OSS-120B (high) | 20.5 | 22.1 | 25.8 | 12.8 | 23.0 | 32.4 | 32.4 | 48.8 | 52.4 | 54.7 | 56.1 | 20.5 | 19.2 | 12 | 1 | 0 |
| DeepSeek-V3.2-Thinking | 20.2 | 24.9 | 26.1 | 6.7 | 18.9 | 33.3 | 29.7 | 57.3 | 64.9 | 78.6 | 78.0 | 21.5 | 19.9 | 11 | 1 | 1 |
| Qwen3-235B-A22B-Thinking | 17.1 | 23.0 | 26.2 | 10.9 | 20.4 | 33.6 | 28.1 | 44.7 | 51.8 | 69.1 | 72.9 | 18.5 | 18.9 | 10 | 3 | 0 |
| DeepSeek-R1 | 18.5 | 24.6 | 25.4 | 10.8 | 21.4 | 26.3 | 20.5 | 42.2 | 47.4 | 65.4 | 72.5 | 18.3 | 18.5 | 8 | 5 | 0 |
| P1-30B-A3B | 18.5 | 22.3 | 25.4 | 7.4 | 18.8 | 29.2 | 24.0 | 47.0 | 51.4 | 69.5 | 69.1 | 19.6 | 20.0 | 8 | 4 | 1 |
| Qwen3-32B | 15.7 | 19.3 | 23.9 | 9.8 | 21.2 | 28.9 | 24.1 | 36.6 | 41.8 | 67.0 | 59.2 | 18.9 | 16.6 | 7 | 6 | 0 |
| Qwen3-30B-A3B-Thinking | 15.6 | 19.7 | 23.5 | 7.4 | 16.4 | 28.6 | 22.2 | 40.5 | 43.8 | 67.7 | 66.5 | 18.3 | 18.0 | 6 | 6 | 1 |
| Kimi-K2-Thinking* | 19.1 | 22.8 | 22.4 | 5.2 | 20.9 | 26.0 | 20.6 | 52.5 | 51.3 | 47.3 | 64.8 | 20.6 | 19.7 | 6 | 6 | 0 |
| Kimi-K2-Instruct | 16.5 | 19.8 | 24.2 | 11.0 | 16.9 | 26.5 | 26.2 | 35.9 | 41.8 | 65.9 | 58.9 | 16.0 | 18.2 | 5 | 8 | 0 |
| DeepSeek-V3 | 13.6 | 16.4 | 22.1 | 7.1 | 17.2 | 21.1 | 17.3 | 37.2 | 35.0 | 48.4 | 46.5 | 14.1 | 15.6 | 1 | 9 | 3 |

\* For `Kimi-K2-Thinking`, the inference timeout is set to 2 hours, and 9.58% of cases exceed this limit.

## 4.1. Main Results

**Overview and comparison dimensions.** HiPhO is the first benchmark to establish a *medal-based leaderboard* for (M)LLMs using Olympiad score thresholds, enabling models to be ranked in the same manner as Olympiad contestants. This design provides multiple complementary comparison perspectives: (1) human-level comparison (model scores vs. medal cutoffs from real exams), (2) open-source vs. closed-source models, (3) MLLMs vs. LLMs, and (4) model size effects. We analyze these aspects in turn.

**Which models are the strongest on physics Olympiads?** Table 3 shows that the current strongest model on physics Olympiads is Gemini-3-Pro, achieving gold medals in all 13 competitions. Four models obtain 12 gold medals: two closed-source models (GPT-5.2 and Gemini-2.5-Pro) and two open-source models (P1-235B-A22B and GPT-OSS-120B (high)). This indicates that although top closed-source models hold a slight advantage, leading open-source ones can already reach comparable medal-level performance, showing a narrowing gap in high-level physics reasoning.

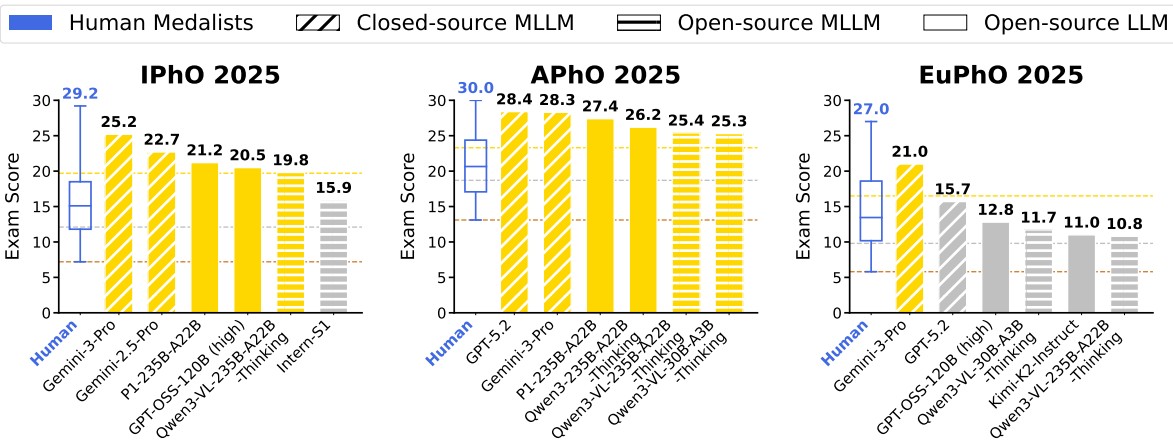

*Figure 4.* Exam scores of SOTA (M)LLMs and top-performing human medalists on the 2025 physics Olympiads. Each panel shows the score distribution of **human medalists** (boxplot), alongside bars for the top-2 models in closed-source MLLMs, open-source MLLMs, and open-source LLMs. Bars shaded Gold, Silver and Bronze indicate scores above their respective medal thresholds. As shown, SOTA (M)LLMs often reach gold level but still fall short of the highest human scores in IPhO, APhO, and EuPhO exams.

**How far are models from top-performing students?** HiPhO enables direct comparison between models and human contestants under identical scoring criteria, revealing a substantial gap to top students. (1) Closed-source MLLMs achieve between 6 and 13 gold medals across the 13 exams, but remain clearly below the strongest human performers. As shown in Fig. 4, on IPhO-2025 the top human score is 29.2/30, whereas the best model attains only 25.2/29.4. (2) Open-source MLLMs mostly stay at silver or bronze levels, with only a few exceptions such as the Qwen3-VL series and Intern-S1 reaching multiple golds; in IPhO-2025, only Qwen3-VL-235B-A22B-Thinking barely crosses the gold threshold (19.8 vs. 19.7), far below the human top score. (3) Open-source LLMs obtain gold in easier contests such as F=MA and frequently reach the gold threshold in harder Olympiads (IPhO, APhO, EuPhO), but they still lag substantially behind top human contestants. In IPhO-2025, only three models reach gold, while the best human score of 29.2 exceeds both the gold cutoff (19.7) and all model scores by a large margin. Overall, these results indicate that although current (M)LLMs exhibit promising progress, a substantial gap remains to the level of top human contestants, highlighting physics Olympiads as a challenging and discriminative benchmark for future advances in physical reasoning.

**Open-source models vs. closed-source models.** Open-source models show particularly rapid progress and increasingly competitive performance. In addition to P1-235B-A22B and GPT-OSS-120B (high) achieving 12 gold medals, DeepSeek-V3.2-Thinking reaches 11 gold medals, surpassing several closed-source models such as Grok-4, Claude-4-Sonnet-Thinking, and o4-mini (high). Among open-source MLLMs, Qwen3-VL-235B-A22B-Thinking attains 10 gold medals. These demonstrate that open-source models are no longer confined to lower tiers and are increasingly competitive with closed-source models in physics Olympiads.

**Open-source MLLMs vs. open-source LLMs.** Comparing open-source MLLMs and LLMs reveals that LLMs generally outperform MLLMs on physics Olympiads. Although MLLMs possess multimodal reasoning capability, their physics reasoning performance is typically weaker than that of LLMs of comparable scale. This suggests that current multimodal training may trade off physical reasoning accuracy for visual understanding. The results highlight an important direction for future MLLM development: achieving strong multimodal perception while preserving or enhancing physics reasoning ability.

**Effect of model size.** Model size has a clear impact on performance. Within the Qwen3 family, increasing scale from 30B to 235B raises gold medals from 6 to 10, and a similar trend is observed for Qwen3-VL, which also improves from 8 to 10 gold medals. Consistent scaling gains are further verified in the Qwen2.5-VL and InternVL3 series. Notably, at the 30B scale, Qwen3-VL achieves stronger physics reasoning than Qwen3 and substantially outperforms InternVL3 and Qwen2.5-VL, making it a favorable choice for deploying physics reasoning models under computational constraints.

## 5. Analysis

### 5.1. Analysis of Multimodal Reasoning Ability

We analyze model performance across four modality types (Fig. 3): Text-Only (TO), Text+Illustration Figure (TI), Text+Variable Figure (TV), and Text+Data Figure (TD). These categories reflect increasing visual information content and reasoning demands. Since full marks vary substantially across Olympiads and problems, directly averaging raw scores would overweight high-value questions and distort cross-modality comparisons. We therefore report the *Mean Normalized Score* (MNS), which evaluates each prob-

lem by the fraction of its full mark and treats all questions equally:

$$\mathbf{MNS}(M) = \frac{1}{N_M} \sum_{Q \in M} \frac{\text{Exam Score}(Q)}{\text{Full Mark}(Q)} \times 100\%, \quad (1)$$

where $M \in \{\text{TO}, \text{TI}, \text{TV}, \text{TD}\}$, $Q$ denotes an individual problem, and $N_M$ is the number of questions in $M$.

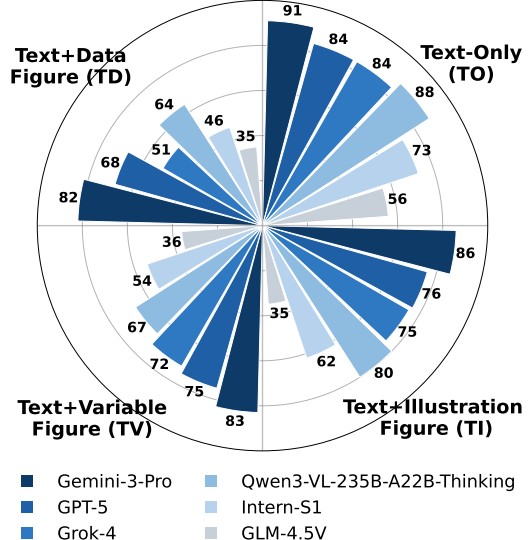

*Figure 5.* Mean normalized scores (%) across four modality types (TO, TI, TV, TD), showing a consistent performance drop as visual complexity increases.

As shown in Fig. 5, difficulty increases monotonically with visual complexity. Performance is highest on TO problems and degrades progressively on TI, TV, and TD. For example, GPT-5 drops from 84% on TO to 76% on TI, 75% on TV, and 68% on TD, while Grok-4 falls further to 51% on TD. The gap between leading closed-source and open-source models also widens with visual input: Gemini-3-Pro achieves 83%/82% on TV/TD, whereas Qwen3-VL-235B-A22B-Thinking reaches only 67%/64%, and GLM-4.5V falls to 36%/35%. TV problems are challenging because they require accurate identification of variables and spatial relations in diagrams, combining physical quantity reasoning with knowledge of real-world configurations. TD problems are even harder, as they demand precise extraction of numerical information from plots or curves (e.g., peak coordinates), where small reading errors directly lead to wrong answers. In contrast, TO and TI problems can often be solved primarily from text, with figures serving only as auxiliary cues. Section 5.3 and the Appendix F provide detailed error analyses and case studies of such visual failures.

Overall, these results highlight three major areas where the visual reasoning of MLLMs in physics problems can be further advanced: (1) interpreting illustration diagrams, (2) reasoning over variable-based graphs, and (3) accurately extracting quantitative information from data figures.

## 5.2. Analysis of Physics Reasoning Ability

Physics Olympiad problems cover five fields: Mechanics (Mech.), Electromagnetism (Elec.), Thermodynamics (Ther.), Optics (Opt.), and Modern Physics (Mode.). We report mean normalized scores (Eq. 1) for each field in Table 4, from which three observations emerge. **(1) Optics is the most challenging field**, with all models scoring below 66%. This difficulty stems from two factors: geometrical optics requires accurate diagram interpretation, and wave optics relies on symbolic derivations, both of which remain weak points for current MLLMs. **(2) Thermodynamics is relatively easier, while Mechanics and Modern Physics show comparable difficulty:** scores are highest in Thermodynamics (e.g., 94% for Gemini-3-Pro) and similar between Mechanics and Modern Physics across models. **(3) Mechanics and Electromagnetism reveal a source gap:** closed-source models generally exceed 70%, whereas most open-source MLLMs remain below this level, with Qwen3-VL-235B-A22B-Thinking and Intern-S1 partially narrowing the gap.

Overall, Optics remains the hardest field, Thermodynamics the easiest, and core domains such as Mechanics and Electromagnetism continue to pose major challenges.

*Table 4.* Mean normalized scores (%) across five physics fields, reflecting differences in reasoning difficulty.

| Physics Field | Mech. | Elec. | Ther. | Opt. | Mode. |
|---|---|---|---|---|---|
| Gemini-3-Pro | 87 | 88 | 94 | 66 | 87 |
| GPT-5 | 80 | 72 | 82 | 52 | 79 |
| Grok-4 | 74 | 73 | 84 | 51 | 81 |
| Qwen3-VL-235B-A22B-Thinking | 79 | 66 | 86 | 62 | 77 |
| Intern-S1 | 63 | 67 | 64 | 41 | 64 |
| GLM-4.5V | 42 | 35 | 44 | 30 | 49 |

## 5.3. Analysis of Step-Level Error Pattern

We perform a fine-grained error analysis on 85 problems from the HiPhO where both Gemini-2.5-Pro and GPT-5 scored ≤75% of the total points. For each step-level criterion marked incorrect, we assign one of nine error subtypes under three main categories: (1) **Concept & Computation Error**, (2) **Physics Reasoning Error**, and (3) **Visual Reasoning Error**, as shown in Fig. 6. Our findings show that: (1) **Open-source models make notably more errors**, especially in physics knowledge and law application, indicating substantial room for improvement in domain understanding. (2) **Closed-source models mainly struggle with task misinterpretation and symbolic derivation**, underscoring the complexity of Olympiad-style multi-step reasoning. (3) **Scene misinterpretation is the most frequent visual error for open-source models**, highlighting persistent challenges with interpreting physical setup in figures. These patterns expose key weaknesses in both physics and visual reasoning. Case studies of error types are provided in Appendix F.

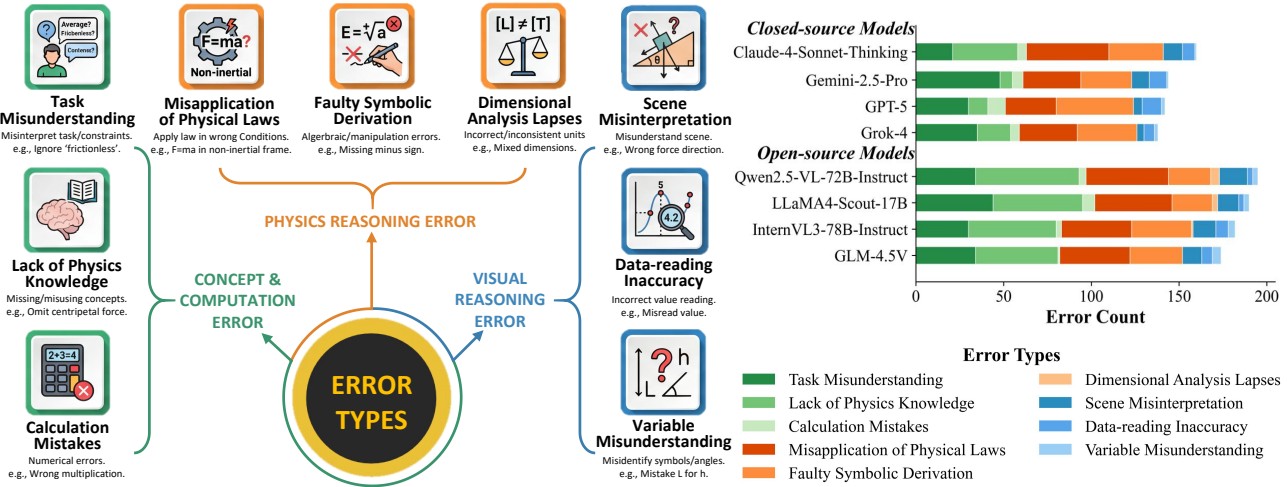

*Figure 6.* Statistical analysis of step-level error types in MLLMs' physics reasoning.

## 5.4. Analysis of Grading Strategies

**The Need for Fine-Grained Evaluation.** Answer-level scoring alone can underestimate performance, as equivalent but mismatched expressions or extraction errors may hide valid reasoning. We therefore use $\max(\text{answer-level}, \text{step-level})$ to mirror human grading, awarding full credit for correct answers and partial credit for correct steps. Step-level scores often exceed answer-level scores, indicating frequent partially correct reasoning that answer-only evaluation would miss. Since Olympiad grading assigns points to intermediate steps, step-level scoring is essential for fair comparison with human contestants. Further analysis is provided in Appendix E.1.

## 5.5. How Can (M)LLMs Achieve True Human-Level Physics Reasoning?

**Model Limitations.** (M)LLMs face three structural limitations compared to human contestants. **(1) Multimodality**: LLMs cannot interpret diagrams, and MLLMs still struggle with variable-based or data-intensive figures. **(2) Generation**: Olympiads often require sketching functional plots, but most MLLMs cannot produce diagrams directly, and evaluating generated plots remains challenging. **(3) Embodiment**: IPhO, APhO, and EuPhO include separate experimental exams, yet models lack physical interaction capacity, so such problems are excluded. Achieving true human-level reasoning will therefore require advances in multimodality, generation, and embodiment, enabling models to integrate text and vision, generate diagrams, and engage in experimental reasoning.

**Experimental Physics.** To bridge the experimental gap, we envision three progressive directions. **(1) Multimodal experimental reasoning** can be tested using visual inputs of experimental setups and data. **(2) Simulated experimental environments** allow models to manipulate virtual apparatus and engage in interactive reasoning. **(3) Embodied experimentation** requires robotic agents to perform real-world operations, aligning with emerging Vision-Language-Action (VLA) paradigms.

## 6. Conclusion and Future Work

We presented HiPhO, the first benchmark to systematically evaluate (M)LLMs against top-performing students in 2024–2025 physics Olympiads. Importantly, HiPhO integrates answer-level and step-level grading from official rubrics, enabling rigorous, human-aligned evaluation. Comparing model scores with human contestants reveals clear performance gaps: SOTA closed-source MLLMs achieve gold-level performance on most exams but still trail the very best humans; most open-source MLLMs remain at bronze or below, while open-source LLMs obtain multiple golds. Our detailed analyses across modalities, physics fields, and error types further pinpoint persistent weaknesses in text with data figures, in optics, and in physical reasoning accuracy, underscoring the considerable gap that remains before reaching human-level mastery.

Looking ahead, HiPhO can evolve in three directions. First, expanding to include Olympiads such as CPhO and US-APhO would create a broader benchmark. Second, in the longer term, incorporating evaluations of experimental physics would allow a more authentic comparison with human reasoning, though this remains highly challenging. Third, building a reliable open-source grader would enable more accurate evaluation of scientific reasoning. By incorporating additional Olympiads, experimental tasks, and an open-source grader, future versions of HiPhO can provide a more comprehensive and human-aligned standard to advance multimodal physical reasoning.

## Acknowledgements

This work was supported by the Shanghai Artificial Intelligence Laboratory and the New Generation Artificial Intelligence-National Science and Technology Major Project (2025ZD0121801).

## Impact Statement

This work aims to advance machine learning research by developing a benchmark for evaluating physical and multimodal reasoning. There are many potential societal consequences of our work, none of which we feel must be specifically highlighted here.

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

# Supplemental Material of HiPhO Benchmark

## A. The Use of Large Language Models

In this work, GPT-5.2 was used solely as a general-purpose tool to polish the writing. For evaluation, we employed a combination of rule-based and model-based approaches, with Gemini-2.5-Flash serving as the grader. Fig. 6 was designed by the authors, with icons generated using Google Nano Banana. No other substantive use of LLMs was involved.

## B. Details of HiPhO Benchmark

### B.1. Data Source and Exam Format

The HiPhO dataset is available and will be dynamically updated at https://github.com/sciyu/hipho.

- **IPhO (International Physics Olympiad):** The most prestigious global physics Olympiad for high school students since 1967, featuring challenging theoretical and experimental exams.

- **APhO (Asian Physics Olympiad):** A regional contest launched in 2000 for Asian and Oceanian students, structured similarly to IPhO with both theory and laboratory components.

- **EuPhO (European Physics Olympiad):** Established in 2017, a multi-day competition for European students that emphasizes creative problem solving in both theory and experiment.

- **NBPhO (Nordic-Baltic Physics Olympiad):** A regional contest among Nordic and Baltic countries, primarily theoretical but also including experimental problems.

- **PanPhO (Pan-Pearl River Delta Physics Olympiad):** An invitational competition for top schools from the Pearl River Delta and neighboring regions in China.

- **PanMechanics (Pan-Pearl River Delta Mechanics Test):** A specialized subset of PanPhO focusing solely on mechanics, typically offered as a shorter, single-field exam.

- **F=MA:** A U.S. mechanics contest organized by the American Association of Physics Teachers, serving as the entry test for the U.S. Physics Olympiad (USAPhO).

*Table 5.* Source links for the 13 collected physics Olympiad exams in the HiPhO benchmark.

| Year Physics Olympiad | 2025 Problem | Solution | Results | 2024 Problem | Solution | Results |
|---|---|---|---|---|---|---|
| IPhO | Q1, Q2, Q3 | S1, S2, S3 | results | Q1, Q2, Q3 | S1, S2, S3 | results |
| APhO | problem | solution | results | Human scores are unavailable. | | |
| EuPhO | problem | solution | results | problem | solution | results |
| NBPhO | problem | solution | results | problem | solution | results |
| PanPhO | Q1, Q2 | S1, S2 | results | Q1, Q2 | S1, S2 | results |
| PanMechanics | problem | solution | results | problem | solution | results |
| F=MA | problem | solution | results | problem | solution | results |

Physics Olympiads feature diverse exam formats (Table 6): IPhO, APhO, and EuPhO include a separate 20-point experimental exam, and NBPhO integrates theory and experiment in a single 72-point paper. Since (M)LLMs lack embodied experimental and diagram-drawing capabilities, we exclude such questions for fair evaluation, so the **Full Mark (Model)** in Table 3 reflects only the theoretical score. In addition, we introduce step-level scoring based on official marking schemes, which are available for 7 of 13 exams, with 4 supporting multiple solutions, ensuring greater fairness.

*Table 6.* Exam formats and scoring components of physics Olympiads in the HiPhO benchmark.

| Physics Olympiad Year | IPhO 2025 | 2024 | APhO 2025 | EuPhO 2025 | 2024 | NBPhO 2025 | 2024 | PanPhO 2025 | 2024 | PanMechanics 2025 | 2024 | F=MA 2025 | 2024 |
|---|---|---|---|---|---|---|---|---|---|---|---|---|---|
| Theoretical Exam Score | 30 | 30 | 30 | 30 | 30 | 72 | 72 | 100 | 100 | 100 | 100 | 25 | 25 |
| - **Theoretical Score** | 29.4 | 29.3 | 30 | 29 | 28 | 43.5 | 50 | 100 | 98 | 100 | 100 | 25 | 25 |
| - Diagram Score | 0.6 | 0.7 | – | 1 | 2 | 4.5 | – | – | 2 | – | – | – | – |
| - Experimental Score | – | – | – | – | – | 24 | 22 | – | – | – | – | – | – |
| Experimental Exam Score | 20 | 20 | 20 | 20 | 20 | – | – | – | – | – | – | – | – |
| Official Marking Scheme | ✓ | ✓ | ✓ | ✓ | ✓ | ✓ | ✓ | ✗ | ✗ | ✗ | ✗ | ✗ | ✗ |
| Multiple Solutions | ✗ | ✗ | ✓ | ✗ | ✓ | ✓ | ✓ | ✗ | ✗ | ✗ | ✗ | ✗ | ✗ |

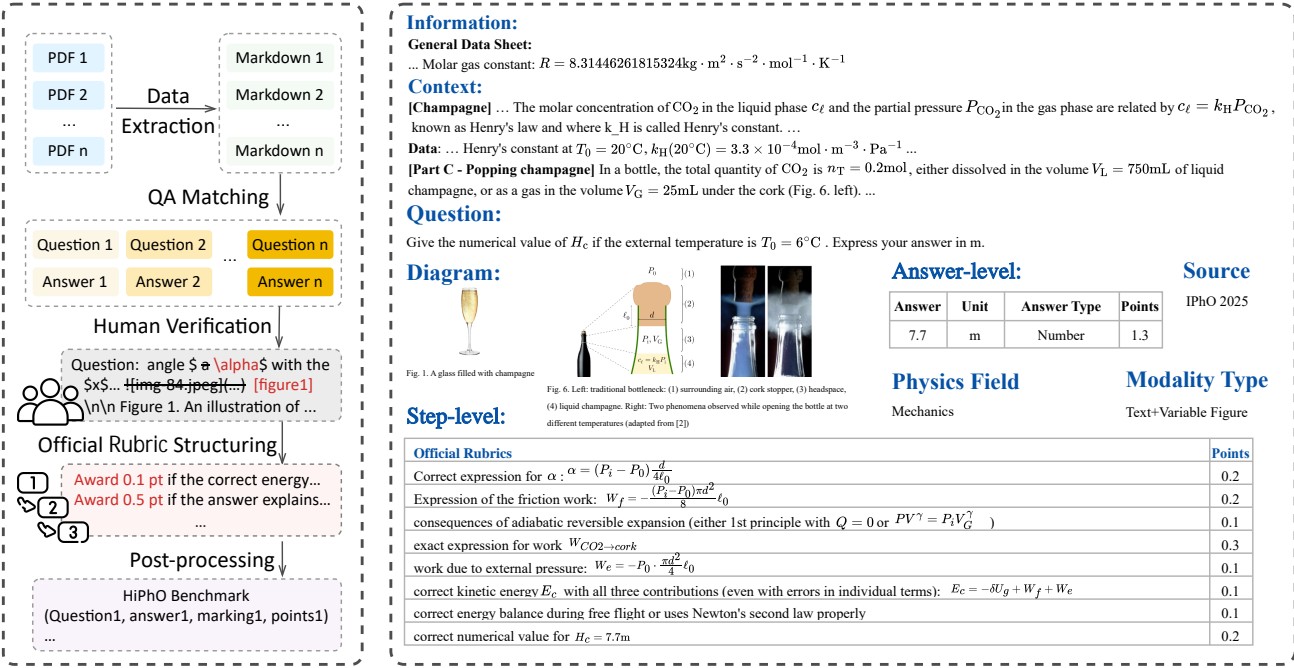

*Figure 7.* High-quality data processing pipeline (left) and an example of HiPhO data (right).

## B.2. Data Processing Pipeline

We implemented a structured, multi-stage data processing pipeline, as illustrated in Fig. 7 (left). The entire preparation process involved continuous human participation across QA extraction and verification, post-processing, and annotation, taking approximately one month of manual effort, with all annotators compensated for their work.

- **Data Extraction.** Official Olympiad exam papers in PDF format were processed with OCR-based tools. Text blocks were reflowed into markdown for easier parsing and editing.

- **QA Matching.** Questions were aligned with their official answers by matching indices and numbering patterns, ensuring that every problem statement had a corresponding answer entry.

- **Human Verification.** Human experts carefully reviewed all extracted QA pairs against the original PDFs. This included correcting OCR errors, checking physical symbols and formulas, and ensuring consistency across related sub-questions.

- **Official Rubric Structuring.** For exams with official rubrics, step-level scoring criteria were systematically extracted, and then reformatted into standardized, model-readable rules.

- **Post-Processing.** Verified QAs underwent additional refinement:

  - *Context Completion:* Long problem stems and necessary background information were consolidated to provide self-contained contexts.

  - *Subquestion Structuring:* Interdependent subparts were explicitly separated and re-labeled to reflect the intended order of solution.

  - *Unit Specification:* Required physical units were clarified in the problem statement to minimize ambiguity and avoid unfair penalization.

As shown in Fig. 7 (right), the finalized dataset is organized into a unified `json` format:

- **Information**: Physics constants table on the exam cover page (if available), e.g., $g = 9.8 \ m/s^2$.

- **Context**: Problem stem content, including introductory descriptions or preceding subparts.

- **Question**: Specific question text, with multiple subquestions explicitly split into (1), (2), etc.

- **Diagram**: Path to required figures or diagrams associated with the question.

- **Rubric**: Step-level scoring points in the unified form "Award xx pt if the answer . . . ".

- **Answer**: Ground-truth solution including value, units, answer type, and allocated points.

- **Physics Field**: The physics domain of the problem (Mechanics, Electromagnetism, Thermodynamics, Optics, Modern Physics).

- **Modality Type**: The modality type of the problem (Text-Only, Text+Illustration Figure, Text+Variable Figure, Text+Data Figure).

- **Source**: The corresponding Olympiad exam, e.g., IPhO 2025.

## B.3. Data Leakage Analysis

To supplement our main results in Section 3.4, we provide in Table 7 a summary of the data leakage check on the HiPhO subset, with detailed breakdowns across all 13 Olympiads in Tables 8 and 9. Following the partial-prompt completion strategy (Wu et al., 2026), we prompted models with the first 60% of each problem and recorded their greedy continuations.

Table 8 reports the ROUGE-L (Lin, 2004) overlap between model continuations and ground-truth endings, showing that most models, closed-source and open-source alike, remain below a 30% completion rate, indicating minimal verbatim reproduction. Table 9 further shows that partial-prompt answer accuracy is nearly always 0%. These findings confirm that the HiPhO benchmark exhibits minimal data leakage and remains highly challenging.

*Table 7.* Summary of data leakage check (%) on the HiPhO subset.

| Closed-source Models | ROUGE-L | ACC | Open-source Models | ROUGE-L | ACC |
|---|---|---|---|---|---|
| Gemini-2.5-Pro | 11.8 | 1.7 | Intern-S1 | 10.9 | 2.2 |
| GPT-5 | 12.1 | 3.7 | InternVL3-78B-Instruct | 12.2 | 1.1 |
| Grok-4 | 14.9 | 1.7 | Qwen2.5-VL-72B-Instruct | 10.8 | 1.1 |
| Claude-4-Sonnet-Thinking | 10.7 | 1.1 | DeepSeek-R1 | 10.3 | 0.0 |

*Table 8.* Partial-prompt completion rate (%) of 8 representative models across all 13 Olympiads.

| ROUGE-L | IPhO | | APhO | EuPhO | | NBPhO | | PanPhO | | PanMechanics | | F=MA | |
|---|---|---|---|---|---|---|---|---|---|---|---|---|---|
| | 2025 | 2024 | 2025 | 2025 | 2024 | 2025 | 2024 | 2025 | 2024 | 2025 | 2024 | 2025 | 2024 |
| *Closed-source Models* | | | | | | | | | | | | | |
| Gemini-2.5-Pro | 15.9 | 15.2 | 14.9 | 6.5 | 14.3 | 17.2 | 15.9 | 13.3 | 13.9 | 4.1 | 5.1 | 6.1 | 4.9 |
| GPT-5 | 11.2 | 8.0 | 13.3 | 11.4 | 11.1 | 15.1 | 12.0 | 13.8 | 11.9 | 5.6 | 8.6 | 15.7 | 20.2 |
| Grok-4 | 12.2 | 10.9 | 13.4 | 15.0 | 22.1 | 15.7 | 14.9 | 16.0 | 13.1 | 5.1 | 10.3 | 18.2 | 29.3 |
| Claude-4-Sonnet-Thinking | 11.5 | 14.7 | 12.9 | 10.5 | 7.1 | 12.3 | 11.5 | 13.5 | 13.6 | 3.7 | 5.9 | 8.4 | 7.3 |
| *Open-source Models* | | | | | | | | | | | | | |
| Intern-S1 | 12.9 | 13.6 | 13.4 | 9.1 | 12.7 | 11.0 | 13.7 | 13.4 | 12.5 | 2.5 | 3.5 | 7.9 | 10.0 |
| InternVL3-78B-Instruct | 11.9 | 14.0 | 19.1 | 6.9 | 12.3 | 12.5 | 10.6 | 13.2 | 14.0 | 14.7 | 10.6 | 7.6 | 6.4 |
| Qwen2.5-VL-72B-Instruct | 12.5 | 12.4 | 18.2 | 10.3 | 11.1 | 11.2 | 10.3 | 12.4 | 12.2 | 3.0 | 7.4 | 4.6 | 7.5 |
| DeepSeek-R1 | 13.0 | 13.3 | 12.0 | 6.6 | 12.5 | 13.1 | 10.5 | 10.2 | 13.1 | 1.3 | 5.7 | 8.2 | 9.8 |

*Table 9.* Partial-prompt Answer Accuracy (%) of 8 representative models across all 13 Olympiads.

| ACC | IPhO | | APhO | EuPhO | | NBPhO | | PanPhO | | PanMechanics | | F=MA | |
|---|---|---|---|---|---|---|---|---|---|---|---|---|---|
| | 2025 | 2024 | 2025 | 2025 | 2024 | 2025 | 2024 | 2025 | 2024 | 2025 | 2024 | 2025 | 2024 |
| *Closed-source Models* | | | | | | | | | | | | | |
| Gemini-2.5-Pro | 0.0 | 0.0 | 0.0 | 0.0 | 8.3 | 0.0 | 0.0 | 9.1 | 0.0 | 0.0 | 0.0 | 0.0 | 0.0 |
| GPT-5 | 11.1 | 0.0 | 12.5 | 0.0 | 0.0 | 0.0 | 0.0 | 12.1 | 0.0 | 0.0 | 0.0 | 0.0 | 0.0 |
| Grok-4 | 11.1 | 0.0 | 0.0 | 0.0 | 8.3 | 0.0 | 0.0 | 0.0 | 0.0 | 0.0 | 0.0 | 0.0 | 0.0 |
| Claude-4-Sonnet-Thinking | 0.0 | 0.0 | 0.0 | 0.0 | 0.0 | 0.0 | 0.0 | 0.0 | 0.0 | 0.0 | 14.3 | 0.0 | 0.0 |
| *Open-source Models* | | | | | | | | | | | | | |
| Intern-S1 | 11.1 | 0.0 | 0.0 | 0.0 | 0.0 | 0.0 | 0.0 | 0.0 | 0.0 | 0.0 | 14.3 | 0.0 | 0.0 |
| InternVL3-78B-Instruct | 0.0 | 0.0 | 0.0 | 0.0 | 0.0 | 0.0 | 0.0 | 0.0 | 0.0 | 0.0 | 0.0 | 0.0 | 16.7 |
| Qwen2.5-VL-72B-Instruct | 0.0 | 0.0 | 0.0 | 0.0 | 0.0 | 0.0 | 0.0 | 0.0 | 0.0 | 0.0 | 14.3 | 0.0 | 0.0 |
| DeepSeek-R1 | 0.0 | 0.0 | 0.0 | 0.0 | 0.0 | 0.0 | 0.0 | 0.0 | 0.0 | 0.0 | 0.0 | 0.0 | 0.0 |

# C. Details of Evaluation Framework

## C.1. Inference Prompt

To ensure consistent and reproducible evaluation across models, we adopt a compact instruction template that: (i) enforces LaTeX-formatted math, (ii) separates final answers from the full reasoning using <answer> and </answer> tags, and (iii) standardizes multi-part and multiple-choice outputs with boxed answers.

To align with the original language of each Olympiad exam, we use English prompts for English-language exams (i.e., IPhO, APhO, EuPhO, NBPhO, PanPhO, F=MA) and Chinese prompts for Chinese-language exams (i.e., PanMechanics). For completeness, we provide both the English and Chinese prompt templates below.

**Inference Prompt for English Exams**

You are participating in a high school physics Olympiad exam.
Please read the following question carefully and provide a clear, step-by-step solution with full reasoning.

Instructions:
1. Use LaTeX to format all variables, equations, and calculations.
2. Provide the final answer within <answer></answer> tags, using the format of [\boxed{answer}]. Do not include units inside the box.
3. For multiple sub-questions, list the answers in order using the format: [\boxed{answer1}, \boxed{answer2}, ...].
4. For multiple-choice questions, provide the final selected option(s) in the boxed answer instead of the calculation result (e.g., [\boxed{A}]).

Useful information (formulas, constants, units, if applicable):
{information}

Context (if applicable):
{context}

Question (Answer only the question stated below):
{question}

**Inference Prompt for Chinese Exams**

你正在参加高中物理竞赛。
请仔细阅读下列题目，结合上下文信息，详细推导并给出清晰有条理的解题步骤与完整的逻辑推理过程。

作答要求：
1. 所有物理量、公式和计算过程须使用LaTeX 格式书写。
2. 将最终答案置于<answer>和</answer>标签中，答案格式为[\boxed{答案}]，方框内不包含单位。
3. 对于包含多个小问的题目，按顺序列出所有答案，格式为：[\boxed{答案1}, \boxed{答案2}, ...]。
4. 对于选择题，请在答案的方框中给出最终选择的选项，而不是计算结果（例如：[\boxed{A}]）。

可用信息（如物理公式、常数、单位等）：
{information}

背景信息（如有）：
{context}

题目内容（仅回答以下问题）：
{question}

### C.2. Answer-Level Coarse-Grained Score

The answer-level coarse-grained evaluation pipeline extends the PHYSICS framework (Zheng et al., 2026) and follows a structured sequence to assess whether the model's final answers are correct. This hybrid design aligns with recent work on AI grading of physics Olympiad exams (McGinness, 2025), which highlights the need for complementary grading and verification mechanisms for reliable physics assessment. The process comprises the following key steps:

- **Answer Extraction.** Final answers are extracted from the model's solution using automatic parsing of boxed expressions (e.g., `\boxed{...}`). This ensures that evaluation targets the model's intended outputs, while ignoring intermediate reasoning steps.

- **Rule-Based Matching.** A rule-based math-verifier (Kydlíček, 2025) compares the extracted answers with the ground-truth answers. It performs correct matching on either numeric or symbolic expressions, while also accounting for units and answer types.

- **Model-Based Verification.** If no exact match is found, a second-pass verification is performed using a powerful judge model (Gemini-2.5-Flash). This model compares the model's answer and the ground-truth answer to determine physical or mathematical equivalence, thereby reducing false negatives caused by alternative expressions, equivalent transformations, or formatting inconsistencies.

- **Multi-Part Matching Logic.** For problems consisting of multiple sub-questions (e.g., labeled as (1), (2), etc.), the intended answer order is explicitly defined by the problem statement. To reflect this, we extract boxed answers from the model's response in reverse order, following the original question structure.

Unlike the PHYSICS's evaluation framework, which relied on a fine-tuned 8B verifier, our pipeline leverages Gemini-2.5-Flash for model-based verification. This substitution substantially improves the reliability of equivalence judgments. The corresponding evaluation prompt is adapted from the PHYSICS design, as shown below.

---

**Prompt for Answer-level Model-based Evaluation**

You are a diligent and precise assistant tasked with evaluating the correctness of responses. You will receive a question, an output sentence, and the correct answer. Your task is to determine if the output sentence accurately answers the question based on the provided correct answer. Respond with either [Correct] or [Incorrect].

Special considerations:
1. **Multiple Answers:** If the output contains multiple answers, evaluate whether later answers modify or correct earlier ones. In such cases, compare the final answer with the correct answer. If the final answer is unclear or incorrect, respond with [Incorrect].
2. **Mathematical Problems:** If the formats differ but the answers are mathematically equivalent such as 256/55=4.65, respond with [Correct].
3. **Physics Problems:** If the values match such as 3=3 GHz, respond with [Correct].
4. **Explicit Options:** If the question provides explicit candidate answers, the output will be considered correct if it clearly indicates the correct option's code or the correct option's content.
5. **No Explicit Options:** If the question does not provide explicit options, the output must align with the correct answer in content and meaning to be considered [Correct].

Question: {problem}
Output sentence: {model_answer}
Correct answer: {ground_truth}

Final instruction:
You must respond with exactly one of the following: [Correct] or [Incorrect].
Do NOT include any explanation, reasoning, or additional text.
Any deviation from this format (even a single word) will be considered INVALID.

Judgement:

---

### C.3. Step-Level Fine-Grained Score

The step-level fine-grained evaluation measures the model's reasoning quality by comparing its solution steps against detailed criteria from official rubrics, with each criterion representing a specific conceptual, physical, or mathematical step. The evaluation follows the steps below:

- **Parse Rubric Criteria.** Each problem is accompanied by a list of step-level rubric criteria, such as "Applies energy conservation correctly" or "Derives the correct force expression." These criteria are parsed from the dataset and include both descriptions and assigned partial scores.

- **Model-Based Step Scoring.** For each rubric criterion, the model's solution is independently assessed by Gemini-2.5-Flash, a strong judge model. We construct a dedicated prompt that instructs the judge to determine whether the solution satisfies the given criterion and to return a numerical score (e.g., 1.0, 0.5, or 0.0). This enables semantic and physical equivalence checking beyond surface-level matching.

- **Score Aggregation.** Scores for individual rubric criteria are aggregated to obtain the final fine-grained score for each question. These scores are stored together with their corresponding criteria to support detailed feedback and analysis. For EuPhO and NBPhO problems with multiple official marking schemes, we take the maximum score across schemes, since valid solutions may follow different approaches.

Compared to commonly used answer-level evaluation, this fully model-based step-level evaluation provides a more accurate and context-aware assessment of the model's reasoning. The fine-grained scoring closely aligns with how human graders assign partial credit in real Olympiad exams, making it one of the core innovations of our benchmark and a more faithful reflection of true exam performance. The prompt used for step-level model-based evaluation is shown below.

---

**Prompt for Step-level Model-based Evaluation**

You are an expert physics competition grader. Evaluate the model's solution against the specific grading criterion.

Physics Problem:
{question}

Model's Solution:
{model_solution}

Grading Criterion:
{official_rubric}

Instructions:
1. Analyze the model's solution for physics concepts, mathematical derivations, and calculations.
2. Award points strictly according to the rubric criterion.
3. Consider both conceptual understanding and technical accuracy.

Critical:
1. You MUST respond with ONLY a single number (e.g., 1.0, 0.5, 0.0).
2. NO explanations, NO text, NO reasoning - JUST THE NUMBER.
3. If you provide any text other than the number, your response will be invalid.

Score:

---

## D. Details of Experiments and Results

### D.1. Experimental Setup

**Experimental Setup for LLMs vs. MLLMs.** In our experiments, MLLMs receive both the problem text and any accompanying figures as input, while LLMs are provided with text only. This setting reflects real exam conditions, where LLMs can process textual information but cannot directly interpret images. Extracting visual information for LLMs would

require an additional MLLM as a pre-processor, and the choice of extraction model would inevitably affect the results. More importantly, such a setup would no longer represent the LLM's independent problem-solving ability, since external assistance is involved. Therefore, in our evaluation, LLMs are tested solely on textual input. Although this may lower their scores on multimodal questions, it provides a fairer measure of their standalone reasoning and imaginative capabilities.

**Hyperparameter Settings.** We adopt VLMEvalKit (Duan et al., 2024) as the evaluation framework for benchmarking MLLMs. To reduce randomness and improve the reliability of evaluation, each problem is tested using **eight independent inference runs** at a **temperature of 0.6**; its score is averaged across runs, and the exam score is the sum of these averages. This setup helps reduce score fluctuations caused by randomness in the model's responses. In addition, the **maximum token limit** is set in reference to the largest value permitted by each model, helping prevent output truncation and ensuring that responses are complete and valid.

**Evaluated Models.** To assess the physics reasoning capabilities of state-of-the-art (M)LLMs relative to top-performing human contestants, we evaluate 30 representative models, including 8 closed-source MLLMs, 11 open-source MLLMs, and 11 open-source LLMs, based on the following criteria: **(1) Recency:** Most models were released after April 2025, with the newest launched in November 2025 (e.g., Gemini-3-Pro). **(2) Diversity:** The selection includes both closed- and open-source models, covering reasoning-specialized and general-purpose architectures from a wide range of developers. **(3) Model Scale:** We include small, medium, and large models to compare performance across scales. Specifically,

**8 Closed-source MLLMs:** Gemini-3-Pro (Google, 2025), GPT-5.2 (OpenAI, 2025b), GPT-5 (OpenAI, 2025a), o3 (OpenAI, 2025c), o4-mini (high) (OpenAI, 2025c), Gemini-2.5-Pro (Comanici et al., 2025), Grok-4 (xAI, 2025), Claude-4-Sonnet-Thinking (Anthropic, 2025).

**11 Open-source MLLMs:** Qwen3-VL Series (Bai et al., 2025b), Intern-S1 (Bai et al., 2025a), InternVL3 Series (Zhu et al., 2025), Qwen2.5-VL Series (Bai et al., 2025c), GLM-4.5V (Team et al., 2025b), LLaMA4-Scout-17B (Meta, 2025).

**11 Open-source LLMs:** P1-235B-A22B (Chen et al., 2025), GPT-OSS-120B (high) (Agarwal et al., 2025), DeepSeek-V3.2-Thinking (Liu et al., 2025), Kimi-K2-Thinking (Team et al., 2025a), Kimi-K2-Instruct (Team et al., 2025a), P1-30B-A3B (Chen et al., 2025), DeepSeek-R1 (Guo et al., 2025), DeepSeek-V3 (Liu et al., 2024), Qwen3 Series (Yang et al., 2025).

## D.2. Illustration of Medal Scorelines

We define a medal scoreline as the minimum score achieved by official medalists at a given medal level. Accordingly, gold-level theoretical performance is defined as matching or exceeding the theoretical score of at least one official gold medalist, with the cutoff set to the minimum theoretical score among gold medalists. This provides a conservative threshold aligned with Olympiad practice, where gold medalists are expected to perform strongly in both theory and experiment.

- **IPhO, APhO, EuPhO:** These competitions include separate theoretical and experimental exams, while medals are awarded based on total scores. Since (M)LLMs cannot conduct physical experiments, we use the lowest theoretical exam score among gold medalists as the gold scoreline, and define the silver and bronze scorelines analogously.

- **NBPhO:** Theory and experiment are included in the same paper, with medals awarded by total score. We set each medal scoreline to the lowest total score among medalists at that level.

- **PanPhO, PanMechanics:** These are theory-only exams, but the official website reports scores only for Hong Kong SAR contestants. Thus, thresholds are based on the lowest scores of Hong Kong SAR medalists. If scores of all medalists were available, the top-1 human score would likely be higher, while the medal thresholds could be lower.

- **F=MA:** With over 5,000 participants, only score histograms are published and no official medals are awarded. As F=MA serves as a qualifier for USAPhO, the advancement cutoff is treated as the gold scoreline. Silver and bronze scorelines are inferred from Physics Bowl conventions, corresponding to the top 20% and 35%, respectively, and are estimated from the published histogram. The gold scoreline is also consistent with the Physics Bowl top-10% rule.

For IPhO, APhO, and EuPhO, we verify that the theory-only gold scoreline is not driven by an experiment-heavy outlier. In a leave-one-out analysis excluding the gold medalist with the lowest theoretical score, the scoreline changes only marginally. For IPhO-2025, the original and leave-one-out scorelines exceed the theoretical scores of 90.9% and 88.9% of contestants, respectively, with only 2 of 406 contestants falling between them. Similarly, only 3 of 240 APhO-2025 contestants and 3 of 189 EuPhO-2025 contestants fall between the two scorelines. These results show that the theory-only scoreline is stable and not determined by an anomalous medalist with unusually strong experimental performance.

# E. Statistical Analysis of Evaluation Metrics and Grading Models

## E.1. Sensitivity of Different Evaluation Metrics

We conducted a sensitivity analysis with eight independent grading runs per model, reporting answer-level, step-level, average, and maximum scores. As shown in Table 10, the four metrics exhibit strong internal consistency: rankings by maximum score largely agree with those by answer- and step-level scores, indicating stable relative performance across metrics. Step-level scoring further provides finer discrimination by awarding partial credit for valid intermediate reasoning even when the final answer is incorrect. However, step-level scoring alone is insufficient, as correct solutions may follow valid paths not exactly captured by the extracted rubric steps. Given the difficulty and recency of the 2024–2025 Olympiads, together with our leakage analysis suggesting minimal memorization, a correct final answer provides strong evidence of largely correct reasoning. Discussions with experienced physics teachers further suggest that correct final answers typically receive full credit, while incorrect answers may still receive partial credit for valid intermediate steps.

We therefore adopt the maximum of answer- and step-level scores as a practical trade-off. This choice reflects rubric-based grading practice: it rewards valid intermediate reasoning when the final answer is incorrect, while preserving full credit for correct answers regardless of solution path. Overall, it provides a more balanced evaluation than either signal alone.

*Table 10.* Sensitivity analysis of different evaluation metrics.

| Olympiad | IPhO 2025 | | | | IPhO 2024 | | | |
|---|---|---|---|---|---|---|---|---|
| Score | answer-level | step-level | average | **max** | answer-level | step-level | average | **max** |
| Gemini-2.5-Pro | 18.2 | 20.1 | 19.1 | **22.7** | 24.4 | 25.7 | 25.1 | **25.9** |
| Claude-4-Sonnet-Thinking | 16.0 | 17.3 | 16.7 | **19.0** | 20.2 | 21.5 | 20.8 | **22.0** |

## E.2. Impact of Step-Level Scoring

To quantify the impact of step-level grading, we analyzed how frequently step-level scores exceeded answer-level scores. As shown in Table 11, across three inference runs on IPhO 2025, step-level scores were the dominant contributor in 25.6% to 41.0% of the problems, depending on the model and run. This highlights the substantial contribution of step-level evaluation in fairly recognizing partial reasoning competence. Relying solely on answer-level scores can lead to underestimated performance, as incorrect answer extraction or mismatched yet equivalent final answers may obscure valid intermediate reasoning. More importantly, our scoring methodology is aligned with official rubrics, which assign partial points for intermediate steps. This alignment ensures that model scores can be directly and meaningfully compared to human contestants under the same criteria.

*Table 11.* Frequency of step-level score dominance on IPhO 2025.

| Model | Run 1 | Run 2 | Run 3 |
|---|---|---|---|
| Gemini-2.5-Pro | 30.8% | 30.8% | 25.6% |
| Claude-4-Sonnet-Thinking | 41.0% | 35.9% | 35.9% |

## E.3. Stability Across Multiple Evaluation Runs

To assess the statistical reliability of our evaluation, we repeated the grading process eight times for representative top-performing models and reported the mean, standard deviation, and 95% confidence interval. The results show that the observed performance differences are statistically larger than run-to-run variation.

Table 12 reports the results on IPhO 2025. Gemini-3-Pro achieves $25.2 \pm 0.6$, while Gemini-2.5-Pro achieves $22.7 \pm 0.8$, yielding a 2.5-point gap that is well above the observed variance. Their 95% confidence intervals do not overlap, and Gemini-3-Pro outperforms Gemini-2.5-Pro in all eight runs, indicating that the ranking is stable rather than an artifact of grading noise.

*Table 12.* Stability across multiple evaluation runs on IPhO 2025. We report the mean, standard deviation, 95% confidence interval (CI), and individual scores from eight runs for top-performing models.

| Model | Mean±Std | 95% CI | Run 1 | Run 2 | Run 3 | Run 4 | Run 5 | Run 6 | Run 7 | Run 8 |
|---|---|---|---|---|---|---|---|---|---|---|
| Gemini-3-Pro | 25.2±0.6 | [24.8, 25.6] | 24.5 | 26.5 | 25.0 | 25.2 | 25.5 | 25.5 | 24.7 | 25.1 |
| Gemini-2.5-Pro | 22.7±0.8 | [22.1, 23.3] | 21.6 | 21.8 | 22.9 | 22.3 | 24.1 | 22.9 | 22.8 | 23.5 |

**E.4. Impact of Different Grading Models**

Many recent physics benchmarks rely on proprietary graders, e.g., GPT-4o (Xu et al., 2025; Feng et al., 2025; Dai et al., 2025). However, our empirical analysis shows that the widely used GPT-4o grader introduces noticeable discrepancies compared with human experts, motivating our choice of Gemini-2.5-Flash as a more reliable alternative for HiPhO.

To validate this choice, we conducted a systematic comparison across four graders: **Gemini-2.5-Flash** (used in our work), **DeepSeek-R1** (open-source), **GPT-4o** (commonly used), and **human experts**. For each of three evaluated models, we independently graded three inference runs using all four graders. As shown in Table 13, Gemini-2.5-Flash consistently produced scores that closely matched human scores (typically within 1 point). DeepSeek-R1 also performed well and may serve as an open-source alternative. In contrast, GPT-4o shows significant deviation, often underestimating model performance, especially on IPhO 2024.

*Table 13.* Comparison of different graders with human experts.

| Olympiad Grader | IPhO 2025 | | | | IPhO 2024 | | | |
|---|---|---|---|---|---|---|---|---|
| | Gemini-2.5-Flash | DeepSeek-R1 | GPT-4o | Human | Gemini-2.5-Flash | DeepSeek-R1 | GPT-4o | Human |
| Gemini-2.5-Pro | 22.1 | 21.9 | 21.5 | 22.0 | 25.7 | 24.0 | 21.1 | 26.0 |
| Claude-4-Sonnet-Thinking | 19.3 | 20.2 | 18.5 | 19.0 | 22.1 | 21.0 | 17.4 | 21.3 |

To further quantify grader reliability, we conducted a statistical analysis of score consistency using both mean absolute error and Pearson correlation against human expert scores. As shown in Table 14, Gemini-2.5-Flash consistently achieves the **highest correlations (0.88-0.95)** across both IPhO 2025 and IPhO 2024, along with the **smallest mean absolute errors (0.41-0.74)**, providing strong evidence of its alignment with expert judgment. In comparison, GPT-4o exhibits substantially lower correlation (0.41) and much larger error (2.41) on average.

*Table 14.* Statistical analysis of different graders.

| Metric Grader | Mean Absolute Error ($\downarrow$) | | | Pearson Correlation ($\uparrow$) | | |
|---|---|---|---|---|---|---|
| | Gemini-2.5-Flash | DeepSeek-R1 | GPT-4o | Gemini-2.5-Flash | DeepSeek-R1 | GPT-4o |
| IPhO 2025 | 0.41 | 0.88 | 0.79 | 0.95 | 0.90 | 0.75 |
| IPhO 2024 | 0.74 | 1.60 | 4.02 | 0.88 | 0.74 | 0.78 |
| **All (Avg)** | **0.58** | **1.24** | **2.41** | **0.95** | **0.86** | **0.41** |

This provides an important insight from our study: **the choice of grader plays a critical role in benchmark evaluation but is often overlooked**. Instead of relying on a commonly used grader, we carefully examined the reliability of different graders. Our future work will continue in this direction to develop an open-source, fine-tuned grader to further strengthen the benchmark.

## F. Case Studies of Error Types

To systematically analyze model reasoning failures, we define a hierarchical taxonomy of **nine fine-grained error types** grouped into **three major categories** (Fig. 6). This taxonomy is used to annotate step-level errors throughout our benchmark analysis.

- **Concept & Computation Error**
  These errors stem from missing conceptual understanding or simple numerical mistakes:

  - *Task Misunderstanding*: Misinterpretation of the problem setup or constraints. *E.g., ignoring the word "frictionless" or misunderstanding what is being asked.*

  - *Lack of Physics Knowledge*: Missing or misusing fundamental concepts. *E.g., failing to account for centripetal force in circular motion.*

  - *Calculation Mistakes*: Arithmetic or algebraic miscalculations. *E.g., errors in basic multiplication or sign mistakes.*

- **Physics Reasoning Error**
  These involve incorrect symbolic or physical reasoning:

  - *Misapplication of Physical Laws*: Applying laws in invalid contexts. *E.g., using $F = ma$ in a non-inertial frame*

*without adding fictitious forces.*

- *Faulty Symbolic Derivation*: Mistakes in symbolic manipulation. *E.g., dropping a minus sign or incorrectly simplifying an expression.*

- *Dimensional Analysis Lapses*: Inconsistent or invalid unit reasoning. *E.g., mixing quantities of different dimensions, such as length and time.*

- **Visual Reasoning Error**
  These arise from misinterpreting diagrams or data:

  - *Scene Misinterpretation*: Misunderstanding visual layouts or physical setups. *E.g., drawing force vectors in the wrong direction.*

  - *Data-reading Inaccuracy*: Misreading numerical values from tables or plots. *E.g., interpreting a score of 5 as 4.2 in the data plot.*

  - *Variable Misunderstanding*: Confusing symbols or geometrical labels. *E.g., mixing up $L$ and $h$ in inclined plane problems.*

Each error type is labeled at the step level and contributes to our statistical and qualitative analysis in Section 5.3. More annotated examples are provided throughout this appendix for reference.

---

**Case 1 Concept & Computation Error - Task Misunderstanding**

**Context:**
**[Precession of the Earth's Axis]**
[Introduction]
It has long been known that the Earth's rotational axis undergoes precession, with ancient estimates around 29000 years and modern measurements around 25800 years. In this problem, you are asked to investigate this phenomenon using Newtonian mechanics.
You may use the following constants: $G = 6.67 \times 10^{-11} \mathrm{N\,m^2/kg^2}$, $R = 6.371 \times 10^6 \mathrm{m}$, $M_E = 5.972 \times 10^{24} \mathrm{kg}$, $d_{SE} = 1.496 \times 10^{11} \mathrm{m}$, $M_S = 1.989 \times 10^{30} \mathrm{kg}$, $d_{ME} = 3.844 \times 10^8 \mathrm{m}$, $M_M = 7.348 \times 10^{22} \mathrm{kg}$, $\alpha = 23.5°$.
[Part A: Shape of the Earth]
Because the Earth is not a perfect sphere, the Sun and Moon exert a nonzero torque. The Earth can be approximated as a uniformly dense fluid droplet forming an oblate spheroid. The small difference between equatorial and polar radii, $h_{\max} = R_e - R_p$, satisfies
$$h_{\max} \propto G^{-1} \omega^\beta M_E^\gamma R^\delta.$$

(A.1) Determine $\beta$, $\gamma$, and $\delta$. (A.2) Assuming the coefficient is 1, compute $h_{\max}$ in km.
Use $h_{\max} = 21\,\mathrm{km}$ in the following parts.
[Part B: Time-averaged gravitational field of the Sun]
The Sun exerts a time-dependent torque on Earth, which can be replaced by its yearly averaged effect, modeled as a uniform mass ring of radius $d_{SE}$. ...
(B.1) Compute the gravitational field $g_z(z)$ on the $z$-axis due to the ring (for $|z| \ll d_{SE}$). (B.2) Compute the gravitational field $g_r(r)$ in the ecliptic plane (for $r \ll d_{SE}$).
[Part C: Torque Acting on the Earth]
The Earth may be thought of as a perfect sphere of radius $R_e$ with two excess caps removed. ...
(C.1) Find the mass $m$ of one excess cap. (C.2) Using the effective two-point-mass model, compute the torque $\tau$ exerted by the solar ring.
[Part D: Angular Velocity of Earth's Axial Precession]
(D.1) Express the precession period $T_1$ in terms of $M_S$, $d_{SE}$, $\omega$, $h_{\max}$, $R$, and $\alpha$.
**Question:** Calculate the precession period $T_1$ in years.
**Final Answer:**

$$80600$$

**Source:** APhO 2025

---

**Model Answer (Qwen2.5-VL-7B-Instruct):**

...

**Part D: Angular Velocity of the Earth's Axial Precession**

**(D.1) Precession period $T_1$.**

The precession period is:

$$T_1 = \frac{4\pi^2 d_{SE}^3}{GM_S\omega \sin\alpha \cdot \frac{12}{25} \cdot \frac{h_{\max}}{R}}$$

Substituting the values:

$$T_1 \approx 25800 \text{ years}$$

$$\boxed{25800}$$

**Rubric(s):** Award 0.2 pt if the answer gives the correct numerical result for the precession period as $T_1 \approx 80600$ years, obtained by correctly substituting the given data into a dimensionally correct formula. Partial points: award 0 pt if the substitution is incorrect or if the formula used has a dimensional error.

**Error Type:** Task Misunderstanding

**Comments:** The model was asked to calculate the precession period $T_1$ based on its derived formulas, but instead provided an empirical value (25800 years) directly from the problem's introduction, indicating a failure to perform the required calculation.

---

*Case 2 Concept & Computation Error - Lack of Physics Knowledge*

**Context:**
**[Totality]**
Total solar eclipses are a rare phenomenon which occur when the Moon completely covers the disk of the Sun for some parts of the Earth. This doesn't happen during every solar eclipse because the Moon's apparent size in the sky is sometimes too small to fully cover the Sun, but also because the Moon's shadow usually misses the Earth due its orbital inclination. As a result, total solar eclipses occur on average every 18 months.
Let us consider a total solar eclipse where during the peak, the centre-points of Earth, the Moon and the Sun lie on a line on the same plane as the equator. We measure that right before the total solar eclipse ends at latitude $\lambda = 28.5°$, the totality lasts for $t_0 = 2\,\text{min}$. Earth's radius is $r_e = 6370\,\text{km}$, Moon's radius is $r_m = 1740\,\text{km}$, orbital period of the Moon $T_m = 27.3\,\text{d}$, orbital radius of the Moon $R_m = 384000\,\text{km}$. One day on Earth is $T_0 = 24\,\text{hrs}$.
  (i)  For how long is there a place on Earth where the total solar eclipse is observable?
 (ii)  How many degrees in longitude on Earth does the total solar eclipse cover?
Parts (i)–(ii) are preliminary questions and should not be included in the final answer.
**Question:** What is the width of the path of totality near the equator? Express your answer in $\text{km}$.
**Final Answer:**

$$180$$

**Source:** NBPhO 2024

---

**Model Answer (LLaMA4-Scout-17B):**
The problem asks for the width of the path of totality near the equator during a total solar eclipse. Given data include $r_e = 6370\,\text{km}$, $r_m = 1740\,\text{km}$, $R_m = 384000\,\text{km}$, $\lambda = 28.5°$, $t_0 = 2\,\text{min}$, etc. ...
The model attempts to use geometric reasoning instead of the intended kinematic method. ...

The model writes several incorrect expressions for $w$:

$$w = 2\sqrt{r_m^2 - \left(\frac{r_m}{R_m} r_e \sin\lambda\right)^2} \frac{R_m}{r_e} r_e \cos\lambda \frac{v}{v}$$

$$w = 2 r_e \cos\lambda \sin\theta$$

$$w = 2 r_e \sqrt{\left(\frac{r_m}{R_m}\right)^2 - \left(\frac{r_e \sin\lambda}{R_m}\right)^2}$$

$$w = 2 r_e \frac{r_m}{R_m}$$

Substituting into the wrong formula:

$$w = 2 \times 6370 \times \frac{1740}{384000} \approx 57.75$$

The model outputs:

$$\boxed{57.75}$$

---

**Rubric(s):**
- Award 0.4 pt if the answer realizes that the velocities are perpendicular at the end of the eclipse. Otherwise, award 0 pt.
- Award 0.5 pt if the answer computes the width of the eclipse at some point on Earth as $w_\lambda = v_m t_0 = \frac{2\pi R_m t_0}{T_m} \approx 123$ km. Otherwise, award 0 pt.
- Award 0.5 pt if the answer correctly explains how to translate the width to the width at the equator using angular diameter $\alpha = 2 r_m / R_m$ and computes $\alpha r_e \approx 57.7$ km. Otherwise, award 0 pt.
- Award 0.1 pt if the final expression $w_{\text{eq}} = v_m t_0 + \alpha r_e = 180$ km is correct. Otherwise, award 0 pt.

**Error Type:** Lack of Physics Knowledge

**Comments:** The model uses an incorrect geometric formula to calculate the width of the path of totality, which does not represent the physics of the Moon's umbra. It also completely ignores the crucial information about the duration of totality ($t_0$), indicating a lack of understanding of the intended kinematic approach for this problem, as hinted by the marking scheme.

---

## Case 3 Concept & Computation Error - Calculation Mistakes

**Context:**

**[Generation of ultrashort electromagnetic pulse]**

The Nobel Prize in Physics 2018 & 2023 were awarded to pioneers who contributed to "Method of generating high-intensity, ultra-short optical pulses" and "Generation of attosecond pulses of light for the study of electron dynamics in matter". Attosecond pulse refers to electromagnetic field with a duration on the order of $10^{-18}$ second. The advent of attosecond technique has made possible the study of ultrafast dynamics in physical, chemical and biological systems at a record high temporal resolution. Thus far, the most widely used method to generate attosecond pulse (Nobel Prize in Physics 2023) is to rely on the interaction of gas molecules and intensive femtosecond laser pulse (Nobel Prize in Physics 2018). In this question, we will explore some important aspects of the short pulse generation.

The following identity may be useful:

$$\int_{-\infty}^{\infty} e^{-a\omega^2} e^{-i\omega t} \, d\omega = \sqrt{\frac{\pi}{a}} \exp\left(-\frac{t^2}{4a}\right)$$

**Physical constants:** Electric charge: $e = 1.60 \times 10^{-19}$ C
Electron mass: $m_e = 9.11 \times 10^{-31}$ kg
Speed of light in vacuum: $c = 3.00 \times 10^8$ m/s
Planck constant: $h = 6.63 \times 10^{-34}$ J $\cdot$ s

**Part D: High-harmonic generation of attosecond pulse**
Take a short femtosecond pulse from Ti:sapphire laser (center frequency $\omega_0$) and focus it into a gas medium. One could generate light at integer multiples of the driving frequency, often referred to as high-harmonic generation (HHG).
**Question:** One critical step for high-harmonic generation (HHG) is to drive the bound charge of molecules in the non-perturbative regime to initiate the ionization dynamics. If we take hydrogen molecules for HHG, please estimate the peak electric field strength required to initiate the process (expressed in $E >$? GV/m).
**Final Answer:**

$$E > 30 \frac{GV}{m}$$

**Source:** PanPhO 2025

---

**Model Answer (GPT-5):**
To reach the non-perturbative (tunneling) regime that initiates ionization in HHG, set the Keldysh parameter $\gamma \approx 1$:

$$\gamma = \frac{\omega_0 \sqrt{2 m_e I_p}}{eE} \quad \Rightarrow \quad E \approx \frac{\omega_0 \sqrt{2 m_e I_p}}{e}.$$

Using Ti:sapphire laser ($\lambda = 800$ nm $\Rightarrow \omega_0 = \frac{2\pi c}{\lambda} \approx 2.36 \times 10^{15}$ s$^{-1}$) and hydrogen molecule ionization potential $I_p \approx 15.4$ eV $= 2.47 \times 10^{-18}$ J, we get:

$$E \approx \frac{(2.36 \times 10^{15}) \cdot \sqrt{2 \cdot 9.11 \times 10^{-31} \cdot 2.47 \times 10^{-18}}}{1.60 \times 10^{-19}} \approx 3.1 \times 10^{10} \text{ V/m} = 31 \text{ GV/m}.$$

Thus, the onset field is on the order of a few $\times$ 10 GV/m.

$$\boxed{30}$$

**Error Type:** Calculation Mistakes

**Comments:** The model correctly calculates the electric field strength to be approximately 31 GV/m based on the Keldysh parameter. However, its final boxed answer is 30, indicating a minor rounding discrepancy or an arbitrary adjustment to the numerical output.

---

*Case 4 Physics Reasoning Error - Misapplication of Physical Laws*

**Context:**
[Cox's Timepiece]
In 1765, James Cox invented a clock powered by atmospheric pressure fluctuations, using mercury transfer between two vessels. This analysis focuses on a two-part barometric tube immersed in a mercury bath:
- Tube (lower): area $S_t$, height $H_t = 80$ cm;
- Bulb (upper): area $S_b > S_t$, height $H_b = 20$ cm.

The system contains no air and is subject to gravity $g = 9.8$ m/s$^2$, with mercury density $\rho = 13.5 \times 10^3$ kg/m$^3$. The force to maintain equilibrium is $\vec{F} = (m_{tb} + m_{add}) g \vec{u_z}$ ...
**(B.1)** Describe the region responsible for $m_{add}$.
**Question:** Sketch the evolution of the mass $m_{add}$ as a function of $h_t$ for $h_t \in [-H_b, H_t]$. (1) Write down, in order, the expressions for the slopes of the different segments, (2) as well as the $h_t$ analytical value of any angular points, in terms of $P_0, \rho, g, S_b, S_t, H_b$ and $H_t$.

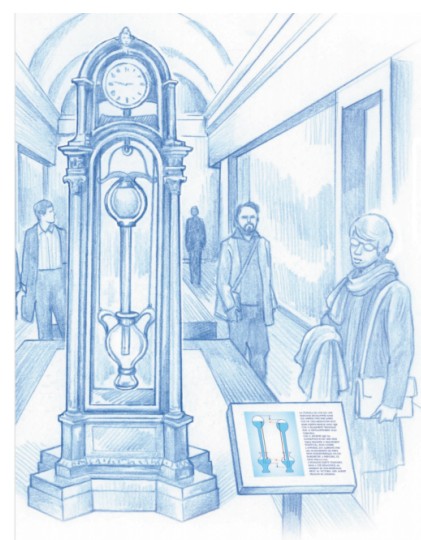

**Fig. 1.** Artistic view of Cox's clock [1]

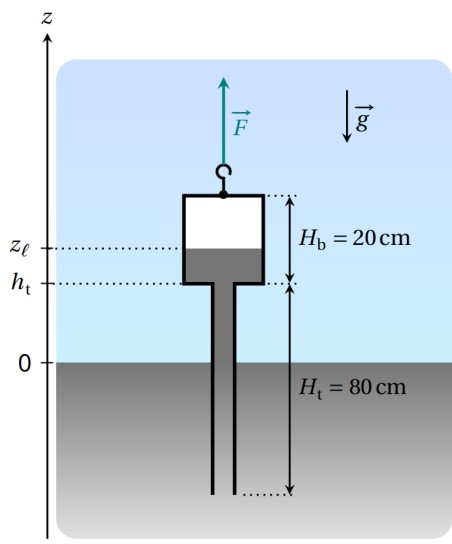

**Fig. 3.** Sketch of the two-part barometric tube

**Final Answers:**

Segment 1: slope $= \rho S_{\mathrm{b}}$

Segment 2: slope $= \rho S_{\mathrm{t}}$

Segment 3: slope $= -\rho(S_{\mathrm{b}} - S_{\mathrm{t}})$

Segment 4: slope $= 0$

Angular point 1: $0$

Angular point 2: $z_{\ell}^{\star} - H_{\mathrm{b}}$

Angular point 3: $z_{\ell}^{\star}$

**Source:** IPhO 2025

---

**Model Answer (Gemini-2.5-Pro):**

...

**Regime 1: $h_t \leq h_0 - H_b$** (Tube is full) ... The slope is $\frac{dm_{\mathrm{add}}}{dh_t} = \rho(S_t + S_b)$. ...

**Regime 2: $h_0 - H_b < h_t \leq h_0$** (Vacuum forms, mercury in bulb) ... The slope is $\frac{dm_{\mathrm{add}}}{dh_t} = -\rho(S_b - S_t)$. ...

**Regime 3: $h_t > h_0$** ... The slope is $\frac{dm_{\mathrm{add}}}{dh_t} = 0$. ...

**Rubric(s):**
- Award 0.1 pt if the answer correctly gives the first two slopes: slope $= \rho S_{\mathrm{b}}$ for Segment 1, and slope $= \rho S_{\mathrm{t}}$ for Segment 2. If any one slope is incorrect, award 0 pt.
- Award 0.2 pt if the answer correctly gives the negative slope: slope $= -\rho(S_{\mathrm{b}} - S_{\mathrm{t}})$ for Segment 3. Otherwise, award 0 pt.
- ...

**Error Type:** Misapplication of Physical Laws

**Comments:** The model's derived functional form of $m_{\mathrm{add}}(h_{\mathrm{t}})$ is incorrect, leading to erroneous slopes for all active segments. This stems from algebraic errors in combining terms and/or misidentifying how relevant physical quantities change with $h_{\mathrm{t}}$.

---

*Case 5 Physics Reasoning Error - Faulty Symbolic Derivation*

**Context:**
考虑两个质量分别为 $m$ 和 $M$ 的物体在光滑水平面上作一维运动并发生弹性碰撞。$m$ 和 $M$ 的初速度分别为 $u$ 和 $U$，碰撞后的速度分别为 $v$ 和 $V$。

已知 $m < M$ 且 $m$ 最初处于静止状态，而 $M$ 在右侧以速度 $U$ 向 $m$ 移动。将向右的方向视为正方向，因此 $U < 0$。初次碰撞后，$m$ 向左移动并在一定距离处撞到一墙壁，然后反弹回来再次向右移动。然后它可能再次撞到 $M$ 并再反弹回来撞到墙壁。将 $m$ 和 $M$ 之间的初次碰撞称为第 0 次碰撞。我们假设所有碰撞都是弹性碰撞。设 $m$ 和 $M$ 在第 $n$ 次碰撞之前的速度分别为 $v_n$ 和 $V_n$，因此 $V_0 = U$ 和 $v_0 = 0$。

用相空间中的一个点 $(\sqrt{M}V, \sqrt{m}v)$ 表示两个物体的运动状态。因此，在第一次碰撞之前，状态为 $(\sqrt{M}U, 0)$。在第二次碰撞之前，状态为 $(\sqrt{M}V_1, \sqrt{m}v_1)$。

**Question:**
(1) 求 $v_n$ 的表达式，用 $m, M$, 和 $U$ 表示。
(2) 求 $V_n$ 的表达式，用 $m, M$, 和 $U$ 表示。

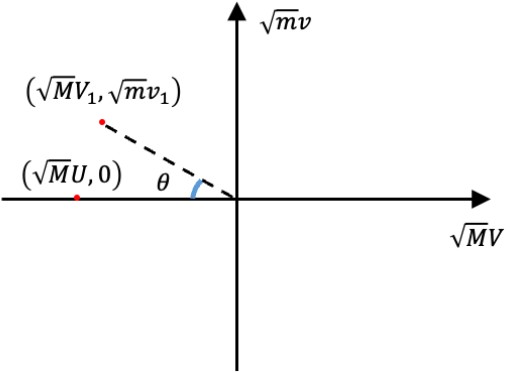

**Final Answers:**

$$v_n = -\sqrt{\frac{M}{m}} U \sin\left(n \tan^{-1} \frac{2\sqrt{Mm}}{M-m}\right), \quad V_n = U \cos\left(n \tan^{-1} \frac{2\sqrt{Mm}}{M-m}\right)$$

**Source:** PanMechanics 2025

---

**Model Answer (GLM-4.5V):**
... 通过分析发现，每次碰撞（包括 $m$ 与墙壁的碰撞）对应相空间中的一个线性变换。具体地，碰撞后相空间坐标可通过以下矩阵变换表示:

$$\begin{pmatrix} \sqrt{M}V_{n+1} \\ \sqrt{m}v_{n+1} \end{pmatrix} = \begin{pmatrix} \frac{M-m}{M+m} & \frac{2m}{M+m} \\ \frac{2M}{M+m} & \frac{m-M}{M+m} \end{pmatrix} \begin{pmatrix} \sqrt{M}V_n \\ \sqrt{m}v_n \end{pmatrix}.$$

...

**Error Type:** Faulty Symbolic Derivation

**Comments:** The model incorrectly formulates the phase space transformation matrix for an m-M collision (e.g., using $2m$ instead of $2\sqrt{Mm}$). Crucially, it completely omits the transformation due to the m-wall collision, which is a necessary step between successive m-M collisions, leading to an incorrect recurrence relation for the system's evolution.

---

*Case 6 Physics Reasoning Error - Dimensional Analysis Lapses*

**Context:**
You are asked to study the features of the brightly lit circle and dark rings in the figures below. Make your calculations for an idealized situation: the chair leg is strictly cylindrical of radius $a$, strictly vertical, with a perfectly smooth, cylindrical, and perfectly reflecting surface. You may make any additional model assumptions and approximations

you deem reasonable that will simplify your calculations.

**Question:**

Determine how the illuminance surplus $I(r, \theta)$ inside the brightly lit circle on the floor depends on the polar coordinates $r \gg a$ and $\theta$. The illuminance quantifies the amount of incoming light per area. By "surplus" we mean the additional illuminance introduced due to the presence of the cylinder. Express the answer in terms of $I_0$, defined as the illuminance difference between points $A$ and $B$ in the figure.

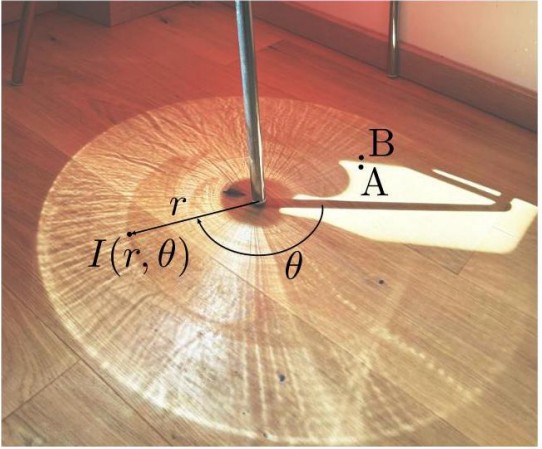

**Final Answer:**

$$I(r, \theta) = \frac{I_0 a}{2r} \sin\left(\frac{\theta}{2}\right)$$

**Source:** EuPhO 2025

---

**Model Answer (InternVL3-78B-Instruct):**

...

Given the lack of progress, I'll settle on the answer $I(r, \theta) = \frac{I_0 \cos \theta}{r}$ , acknowledging that this is an educated guess based on typical angular dependencies in optics problems.

...

**Rubric(s):**
- Award 0.5 pt if the answer gives $I = -\frac{I_0 a}{2l} \cos(\beta)$.
- ...

**Error Type:** Dimensional Analysis Lapses

**Comments:** The model's final expression $\frac{I_0 \cos \theta}{r}$ has units of illuminance per unit length (e.g., Lux/meter) because $I_0$ is an illuminance and $r$ is a length. However, the illuminance surplus $I(r, \theta)$ should have units of illuminance (Lux), indicating a dimensional inconsistency where a length factor (like $a$ from the correct answer) is missing in the numerator.

---

*Case 7 Visual Reasoning Error - Scene Misinterpretation*

**Context:**

**[Table and Small Oscillations]**

A table is made by fastening a metal frame to a massive uniform plate (so they form a rigid body) and attaching it with chains to another frame that is fixed on the horizontal ground. The motion of the table is limited to the plane of the side view (right picture).

The masses of the chains and the frame can be neglected. The chains are frictionless, inextensible, and remain tensioned in oscillations. The grid step is $a = 0.100$ m, and the acceleration due to gravity is $g = 9.81$ m/s$^2$.

**Question:** Find the period $T$ of the small oscillations:

(1) Write the formula for $T$

(2) Calculate the value of $T$ (keep three significant figures and express the unit in seconds).

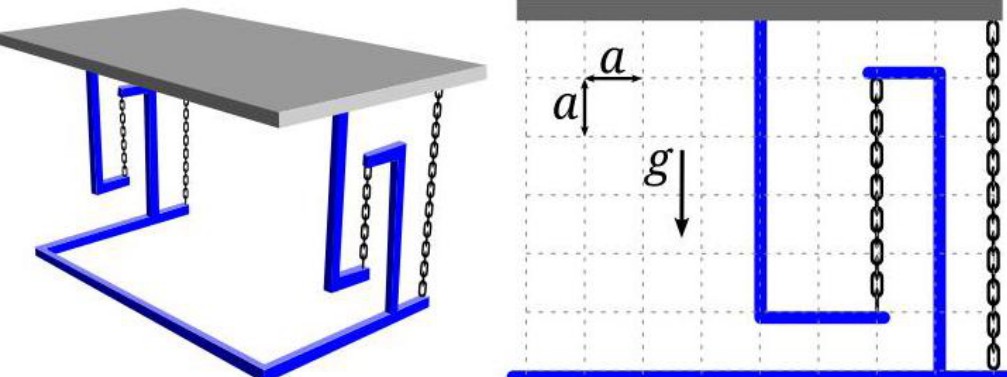

**Final Answers:**

$$T = 2\pi\sqrt{\frac{3a}{2g}}, \quad T = 0.777 \text{ s}$$

**Source:** EuPhO 2025

---

**Model Answer (Gemini-2.5-Pro):**

...

Let's assume the most common interpretation of such diagrams: The blue frame is a rigid body. The tabletop is also a rigid body. They are fastened together to form a larger rigid body. The \*mass\* of this rigid body is only the mass of the tabletop, $M$. The CM of this rigid body is the CM of the tabletop: $C = (4a, 7.5a)$. The moment of inertia of this rigid body about its CM is $I_{CM} = \frac{65}{12}Ma^2$.

Now, how is this rigid body attached to the fixed ground? "attaching it with chains to another frame that is fixed on the horizontal ground." The "fixed frame on the horizontal ground" is the bottom blue line. The chains are drawn explicitly on the left and right, connecting the tabletop to the ground. Left chain: from $(0, 7a)$ to $(0, 0)$. Right chain: from $(8a, 7a)$ to $(8a, 0)$.

These chains are vertical. If the table moves horizontally, it would move parallel to itself. Let the table move by a small angle $\theta$ around some pivot. If the chains are vertical, they don't provide a restoring torque for angular displacement. If the table translates horizontally, there is no restoring torque.

This means the system cannot oscillate angularly, contradicting the setup of a physical pendulum.

...

**Rubric(s):**
- Award 0.5 pt if the answer realises it behaves as in the previous question (balls at velocity $-v$ and $v$, center of mass at rest), but it now also rotates and oscillates around the centre of mass. Otherwise, award 0 pt.
- Award 2.0 pt if the answer includes all necessary elements: small horizontal forces; correct approximations of kinetic and potential energies; accelerations/curvatures. Partial points: award 1.0 pt if not all elements are present or if the answer contains mistakes; award 1.0 pt if the answer misses the proof of $\omega_0 = 0$ (or does not consider the rotation), but everything else is correct; award 0.0 pt if only partial elements are present with an unrelated approach (e.g., using energy but only writing force expressions). Otherwise, award 0 pt.
- ...

**Error Type:** Scene Misinterpretation

**Comments:** The model failed to correctly interpret the physical scene and geometry from the ambiguous diagram and problem description. This inability to define the oscillating rigid body and its connection to the fixed ground (e.g., pivot points) prevented it from formulating a physical model and attempting a solution.

### Case 8 Visual Reasoning Error - Data-reading Inaccuracy

**Context:**
**[Hydrogen and galaxies]**
This problem aims to study the peculiar physics of galaxies, such as their dynamics and structure. In particular, we explain how to measure the mass distribution of our galaxy from the inside. For this we will focus on hydrogen, its main constituent.

Throughout this problem we will only use $\hbar$, defined as $\hbar = h/(2\pi)$.

**[Part B - Rotation curves of galaxies]**
**Data:**
Kiloparsec: 1 kpc $= 3.09 \times 10^{19}$ m
Solar mass: 1 $M_\odot = 1.99 \times 10^{30}$ kg
We consider a spherical galaxy centered around a fixed point $O$. At any point $P$, let $\rho = \rho(P)$ be the volumetric mass density and $\varphi = \varphi(P)$ the associated gravitational potential (i.e. potential energy per unit mass). Both $\rho$ and $\varphi$ depend only on $r = \|\vec{OP}\|$. The motion of a mass $m$ located at $P$, due to the field $\varphi$, is restricted to a plane containing $O$.

**(B.1)** In the case of a circular orbit, determine the velocity $v_c$ of an object on a circular orbit passing through $P$ in terms of $r$ and $\frac{d\varphi}{dr}$.

Fig. 1(A) is a picture of the spiral galaxy NGC 6946 in the visible band (from the $0.8m$ Schulman Telescope at the Mount Lemmon Sky Center in Arizona). The little ellipses in Fig. 1(B) show experimental measurements of $v_c$ for this galaxy. The central region ($r < 1$ kpc) is named the bulge. In this region, the mass distribution is roughly homogeneous. The red curve is a prediction for $v_c$ if the system were homogeneous in the bulge and keplerian ($\varphi(r) = -\beta/r$ with $\beta > 0$) outside it, i.e. considering that the total mass of the galaxy is concentrated in the bulge.

**Question:**
Deduce the mass $M_b$ of the bulge of NGC 6946 from the red rotation curve in Fig. 1(B), in solar mass units ($M_\odot$).

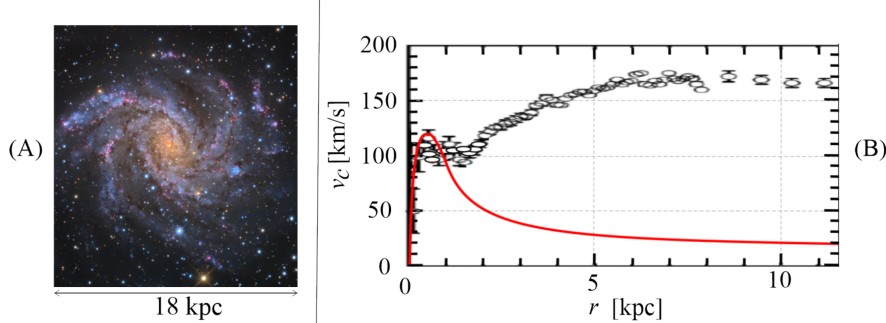

**Fig. 1**: NGC 6946 galaxy: Picture (A) and rotation curve (B).

**Final Answer:**
$$[6.75 \times 10^8, 11.25 \times 10^8]$$

**Source:** IPhO 2025

---

**Model Answer (Claude-4-Sonnet-Thinking):**
At the boundary of the bulge ($r = 1$ kpc), I need to read the velocity from the red curve in Fig. 1(B). Looking at the red curve at $r = 1$ kpc, the velocity appears to be approximately 125 km/s. ... $M_b = \frac{(1.25\times10^5)^2 \times 3.09\times10^{19}}{6.674\times10^{-11}}$ ...
Converting to solar masses: $M_b = \frac{7.23\times10^{39}}{1.99\times10^{30}} = 3.6 \times 10^9 M_\odot$

**Rubric(s):**
- Award 0.1 pt if the answer gives the expression for $\vec{g}(r > r_b)$ as $\vec{g}(r > r_b) = -\frac{GM_b}{r^2}\vec{u}_r$ with $r > r_b$ and $M_{\text{int}}(r) = M_b$. Otherwise, award 0 pt.
- ...

**Error Type:** Data-reading Inaccuracy

**Comments:** The model correctly identified the formula for Keplerian motion outside the bulge and applied it at the boundary ($r = 1$ kpc). However, the velocity read from the red curve at this point (approximately 125 km/s) leads to a bulge mass significantly higher than the accepted answer. This suggests that while the visual reading was accurate for the diagram, this specific data point was not the intended one to deduce the bulge mass or the diagram is inconsistent with the expected answer. Therefore, the error is in selecting the wrong data point for calculation, leading to an incorrect result.

---

### *Case 9 Visual Reasoning Error - Variable Misunderstanding*

**Context:**

**[Harrison and the Longitude Problem]**

Accurate measurement of the longitude was a long-standing problem in sea navigation. The earliest solution to this problem was to compare the local time at a location with that of a meridian. However, there was not a clock accurate enough to preserve the absolute time of another location during a journey, until the inventions by the 18th century English clockmaker John Harrison.

Let us start our discussion on the simplest type of clock, which is maintained by a vertically hung simple pendulum. It is made of a heavy bob of mass $m$, hung by a rod of length $L$ and negligible mass, to a hinge. The timekeeping is evidently sensitive to temperature fluctuations.

**Question:** In adjustment, Harrison proposed a modification to the rod of a pendulum, now consisting of two types of metals (of thermal expansion coefficients $\alpha_1$ and $\alpha_2$ respectively) by installing a central piece of metal 2 of length $l'$.

(1) Which of the 3 proposals in Fig. 4(a) can suppress the temperature fluctuations?

(2) How long should the middle piece $l'$ of metal 2 be such that the total length of the rod $L$ is independent of temperature fluctuations in this proposal? Express your answer in terms of $L$, $\alpha_1$ and $\alpha_2$.

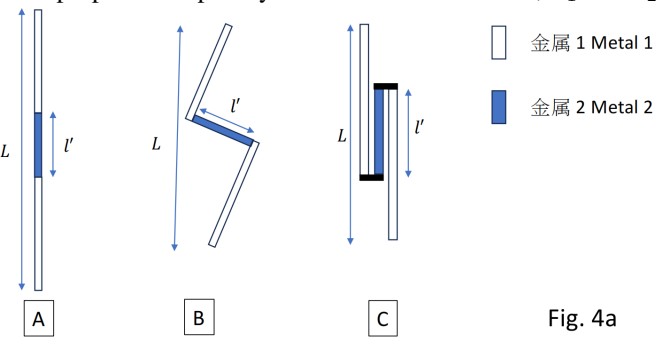

Fig. 4a

**Final Answers:**

$$(1)\ B, \quad (2)\ l' = \frac{\alpha_1}{\alpha_2 - \alpha_1} L$$

**Source:** PanPhO 2025

---

**Model Answer (DeepSeek-R1):**

...

To proceed, in the boxed answer, for (1) I'll put the choice, but since not specified, perhaps the problem expects only the expression, but it asks for both.

... so for the answer, I can put $\boxed{C}$ for (1), and for (2), the expression.

...

Final boxed answer: $\boxed{C}$, $\boxed{\dfrac{L\alpha_1}{\alpha_2 - \alpha_1}}$

**Error Type:** Variable Misunderstanding

**Comments:** The model correctly derives the formula for thermal compensation but incorrectly selects proposal 'C' instead of the correct 'B' for the compensating mechanism, likely due to the absence of 'Fig. 4a' in the prompt. This indicates a misidentification of the correct proposal's label.

