# OpenReview forum: "HiPhO: How Far Are (M)LLMs from Humans in the Latest High School Physics Olympiad Benchmark?"
_ICML.cc/2026/Conference — ICML 2026 regular_

### Official Review · Reviewer_mM6V · 2026-02-20

**Soundness:** 3
**Presentation:** 3
**Significance:** 3
**Originality:** 3
**Overall Recommendation:** 5
**Confidence:** 3

**Summary:**

This paper introduces HiPhO, for high-school physics Olympiads, which compiles 13 recent Olympiad exams from spanning text-only to diagram/data-figure modalities. The benchmark’s uses official rubrics for step-level grading and evaluates 30 state-of-the-art (M)LLMs, reporting medal counts and analyses across modality types, physics fields, and step-level error categories.

**Compliance With Llm Reviewing Policy:**

Affirmed.

**Final Justification:**

My concerns have been adequately addressed.

**Key Questions For Authors:**

Please refer to weakness

**Limitations:**

Please refer to weakness

**Strengths And Weaknesses:**

Strengths

1. The work targets well-motivated shortcomings in prior physics reasoning datasets.
2. TO/TI/TV/TD enables diagnostic analysis of where multimodal models fail.
3. Using official rubrics and translating scores into medal thresholds is meaningful.

Weaknesses
1. Using max(answer-level, step-level) is intuitive to prevent underestimation but without provide a sufficiently rigorous justification that this rule best reflects official Olympiad grading logic? Is there any official guideline or policy? This grading method award partial credit for locally correct derivations. This may reasonable to human evaluation but for a model might/or prompted intended to generate many intermediate steps to increase the chance of receiving partial rubric matches, even if the final answer is incorrect.
2. The paper mentions manual expert verification during dataset construction, but does not provide sufficient details about the verification protocol. e.g.:How many experts were involved? Was there any inter-annotator agreement measurement?

---

> ### Author Rebuttal · Authors · 2026-03-31
>
> We appreciate the reviewer's positive assessment of the motivation, diagnostic design, and rubric-based evaluation in our work. We will further revise the paper accordingly.
>
> ---
>
> **Comment 1.** Using max(answer-level, step-level) is intuitive to prevent underestimation but without provide a sufficiently rigorous justification that this rule best reflects official Olympiad grading logic? Is there any official guideline or policy? This grading method award partial credit for locally correct derivations. This may reasonable to human evaluation but for a model might/or prompted intended to generate many intermediate steps to increase the chance of receiving partial rubric matches, even if the final answer is incorrect.
>
> **Response 1.**  Our metric is intended to balance correctness and reasoning in a way that better reflects real grading practice, and we will clarify this in the revision.
>
> First, it addresses the limitations of answer-only evaluation. Many existing benchmarks assign zero credit when the final answer is incorrect, thereby ignoring partially correct reasoning, especially for Olympiad problems that require multi-step derivations. In contrast, our step-level scoring, grounded in official rubrics, assigns partial credit for valid intermediate steps.
>
> Second, the design is motivated in three considerations. (i) **Step-level rubrics are directly derived from official marking schemes**, with explicit partial credit assigned to each step. (ii) **Answer-level credit is consistent with common exam practice** (e.g., Cambridge STEP: "a candidate reaching the correct answer will receive full marks, regardless of the method used"). (iii) We aim to **approximate common grading practice**: discussions with experienced physics teachers in top-tier high schools confirm that correct final answers are typically awarded full credit, while incorrect answers receive partial credit based on intermediate steps.
>
> Third, we acknowledge the concern that step-level scoring could be exploited by generating excessive intermediate steps due to the characteristics of reasoning models. Actually, this risk is limited because the step-level score, like the answer-level score, is capped. We further mitigate it in three ways: (i) All models are evaluated under **uniform, simple prompts**, without encouraging verbosity. (ii) **Rubrics are derived from official marking schemes**, providing authoritative criteria. (iii) **Rubric matching is selective rather than additive**: only predefined, conceptually correct steps receive credit, so repetitive, irrelevant or incorrect steps do not improve the score; importantly, these rubrics correspond to key logical steps rather than superficial patterns, making it difficult to gain credit through repetitive outputs.
>
> For these reasons, we use max(answer-level, step-level) as a **practical compromise**: it avoids penalizing correct final answers produced through alternative solution paths, while still rewarding valid reasoning when the final answer is wrong. We believe this better reflects how complete and partial solutions are graded in practice.
>
> ---
>
> **Comment 2.** The paper mentions manual expert verification during dataset construction, but does not provide sufficient details about the verification protocol. e.g.:How many experts were involved? Was there any inter-annotator agreement measurement?
>
> **Response 2.** A substantial amount of human effort went into the dataset construction, and we will clarify these in the revision to better highlight both the manual effort and the reliability of high-quality dataset.
>
> As described in Section 3.2, dataset construction involved three stages with human participation throughout: **(1) QA extraction and verification, (2) post-processing, and (3) data annotation**. In total, **9 annotators** with relevant backgrounds, such as former physics competition participants, undergraduate physics students, and graduate students, contributed to these stages. Their work included extracting and verifying QA pairs, conducting post-processing such as explicit sub-question splitting, and annotating metadata such as physics field and visual modality.
>
> For QA extraction and verification, all annotators followed a **consistent verification protocol** to ensured that: (i) content and mathematical symbols were accurately aligned with the original PDFs; (ii) formulas remained coherent across related questions, for example by supplementing formulas when a later sub-question referred to an equation introduced earlier; and (iii) grading rubrics were accurately extracted and refined when the original marking schemes were abbreviated.
>
> Afterward, the processed data were jointly reviewed by **2 annotators with physics competition experience**, who resolved discrepancies through discussion and reached a final consensus. This final review served as cross-validation and a consistency check. Overall, the data preparation process took approximately **one month** of manual effort.

---

> > ### Author Rebuttal · Reviewer_mM6V · 2026-04-02
> >
> > Thank you for your rebuttal. I'll keep my score as it's high enough.

---

> > > ### Author Response · Authors · 2026-04-02
> > >
> > > We greatly appreciate your recognition that our response has fully addressed the concerns, as well as your generous score. Thank you again for your time and thoughtful feedback.

---

### Official Review · Reviewer_HPfG · 2026-03-06

**Soundness:** 3
**Presentation:** 3
**Significance:** 3
**Originality:** 3
**Overall Recommendation:** 4
**Confidence:** 4

**Summary:**

This work introduces HiPho, an up-to-date benchmark consisting of 2024-2025 physics olympiad. The dataset consists of multiple modalities and comes with ground-truth final answers and step-level rubrics. Various open-weight and proprietary models are evaluated, and the results show that the models are already capable of gold-level performance in many hard olympiads.

**Compliance With Llm Reviewing Policy:**

Affirmed.

**Final Justification:**

My questions are mainly answered and I keep my current score.

**Key Questions For Authors:**

In the paper, you mention:
> For each problem, the final score is the maximum of its answerlevel and step-level scores, preserving full credit for correct answers while fairly recognizing valid reasoning.
I am not sure if this is a good metric to capture. You mention that this is how human grade solutions, but don't they also verify the steps of how the solution reached the final answer? I am not familiar with how IPhO is graded, but if the step-level correctness is available, why rely on final answer?

**Limitations:**

Societal impacts are not applicable.

**Strengths And Weaknesses:**

Strengths:
* The problems are from very recent olympiads, which makes contamination less likely.
* The coverage is very comprehensive with rubrics and final answers.
* The evaluation is comprehensive, consisting of many models.
* The paper is generally well-written and easy to follow
Weaknesses:
* There are many other physics olympiad datasets already existing in the prior literature. While the authors focus on the larger number of modalities, step-level accuracy, and lower contamination as their advantage, I am not sure if the conclusions would be different on other datasets (e.g., OlympiadBench is already multi-modal with olympiad level problems). For instance, do the authors find that other datasets are contaminated? (e.g., one model shows high performance on other dataset, while lower performance on their own), or do they find out that step-based grading adds additional signal for better distinguishing models compared to final answers?

Overall, I believe this dataset would be useful to the community as a high-quality less-contaminated source of benchmark, hence I lean towards acceptance.

---

> ### Author Rebuttal · Authors · 2026-03-31
>
> We appreciate the reviewer's positive feedback on the dataset quality, evaluation comprehensiveness, and overall presentation, and we will revise the paper accordingly.
>
> ---
>
> **Comment 1.** There are many other physics olympiad datasets already existing in the prior literature. While the authors focus on the larger number of modalities, step-level accuracy, and lower contamination as their advantage, I am not sure if the conclusions would be different on other datasets (e.g., OlympiadBench is already multi-modal with olympiad level problems). For instance, **(1)** do the authors find that other datasets are contaminated? (e.g., one model shows high performance on other dataset, while lower performance on their own), or **(2)** do they find out that step-based grading adds additional signal for better distinguishing models compared to final answers?
>
> **Response 1.** We agree that prior datasets include Olympiad-level physics problems. HiPhO is designed to complement existing benchmarks along three key dimensions: (i) **up-to-date coverage** (13 Olympiads from 2024-2025 with no temporal overlap with OlympiadBench), (ii) **fine-grained evaluation** using official rubrics with step-level grading, and (iii) **human-aligned comparison** via exam scores and medal thresholds, which are not provided in prior work.
>
> Regarding **(1) contamination**, a direct cross-benchmark comparison with OlympiadBench is currently not feasible. OlympiadBench evaluates earlier-generation models (e.g., GPT-4/4V) on older exams (e.g., IPhO-2021, APhO-2015, EuPhO-2022), whereas HiPhO evaluates recent frontier models on **2024-2025 Olympiads**, with **no overlap in either the dataset or the evaluated models**. Moreover, HiPhO is built from **recent, up-to-date exams** and is carefully curated, with extensive manual verification to ensure high data quality. We also include contamination checks indicating that validation leakage is unlikely, which further reduces concerns about data contamination.
>
> Regarding **(2) step-based grading**, it provides additional and meaningful signal beyond final answers. As shown in **Appendix E.2**, **step-level scores often exceed answer-level scores**, indicating that they capture partially correct reasoning missed by final-answer-only evaluation. For example, on IPhO-2025, step-level scoring improves over answer-level on 25.6%-41.0% of problems across models. This demonstrates that step-based grading offers finer resolution and avoids systematically underestimating model capabilities.
>
> ---
>
> **Comment 2.** I am not sure if this is a good metric to capture. You mention that this is how human grade solutions, but don't they also verify the steps of how the solution reached the final answer? I am not familiar with how IPhO is graded, but if the step-level correctness is available, why rely on final answer?
>
> **Response 2.** Our metric is designed to address the limitations of both answer-only and step-only evaluation, while better approximating rubric-based grading practice.
>
> First, many existing benchmarks rely solely on **answer-level accuracy**, where an incorrect final answer receives zero credit. This ignores partially correct reasoning and can **systematically underestimate model performance**, especially for Olympiad problems requiring multi-step derivations. In contrast, our step-level scoring, based on official rubrics, assigns partial credit for valid intermediate steps and better captures reasoning quality.
>
> However, step-level scoring alone is also insufficient, since a model may reach the correct solution through an alternative path that does not exactly match the extracted rubric steps. Moreover, given the difficulty and recency of the 2024-2025 Olympiads and our leakage analysis suggesting minimal memorization, a correct final answer is strong evidence of largely correct reasoning. Our discussions with experienced physics teachers also suggest that, in practice, correct final answers usually receive full credit, while incorrect answers may still earn partial credit for intermediate steps.
>
> Therefore, we adopt max(answer-level, step-level) as a **practical trade-off** between the two signals. This design aligns with rubric-based grading practices, where (i) partial credit is awarded for intermediate steps, and (ii) correct final answers are credited regardless of the specific solution path (e.g., Cambridge STEP: **"A candidate reaching the correct answer will receive full marks, regardless of the method used"**). It ensures that valid reasoning is rewarded while preserving full credit for correct answers, providing a more balanced evaluation than either signal alone.

---

> > ### Author Rebuttal · Reviewer_HPfG · 2026-04-01
> >
> > Thank you for your rebuttal. My questions are mainly answered and I keep my current score.

---

> > > ### Author Response · Authors · 2026-04-01
> > >
> > > We sincerely thank the reviewer for the positive assessment and for acknowledging that our rebuttal has adequately addressed the concerns. Your recognition and positive feedback are truly valuable to us.

---

### Official Review · Reviewer_83rP · 2026-03-14

**Soundness:** 3
**Presentation:** 3
**Significance:** 3
**Originality:** 2
**Overall Recommendation:** 4
**Confidence:** 4

**Summary:**

HiPhO is a benchmark of 360 physics Olympiad problems drawn from 13 exams across seven competitions from the 2024–2025 cycle, including both text-only and diagram-dependent questions. The benchmark scores models by exam points rather than item accuracy, using answer matching for all problems and rubric-based step grading when official marking schemes are available, then maps scores to medal cutoffs derived from contest results. The paper evaluates 30 LLMs and MLLMs and finds that the strongest closed models and some open LLMs can reach gold level on many exams, but performance drops on figure-heavy problems and remains below the best human contestants.

**Compliance With Llm Reviewing Policy:**

Affirmed.

**Key Questions For Authors:**

1. **Grader family bias.** Did you compare Gemini-family model outputs under Gemini-2.5-Flash and a non-Gemini grader such as DeepSeek-R1? If Gemini models score materially lower under another grader, the leaderboard in Table 3 would need reinterpretation.

2. **Variance across runs.** What are the standard deviations or confidence intervals over the 8 runs for the top models? If differences such as 25.2 vs. 22.7 on IPhO-2025 are within noise, the medal ranking is less convincing.

3. **Medal thresholds for IPhO/APhO/EuPhO.** How many actual medalists fall below your theory-only cutoff, and is the cutoff driven by an outlier with an unusually strong experimental score? If yes, the human-level comparison is less meaningful.

4. **Coverage of rubric-based grading.** Do the main conclusions still hold on the 7 exams with official marking schemes? If not, the pooled leaderboard mixes two different evaluation regimes.

**Limitations:**

yes

**Strengths And Weaknesses:**

The paper addresses a real gap: recent physics benchmarks are either not Olympiad-focused, not multimodal, or not graded with official rubrics. The strongest part is the evaluation design. Step-level grading with official marking schemes is much closer to actual Olympiad assessment than plain answer accuracy, and Table 11 shows that it materially affects scores on a large fraction of problems. The TO/TI/TV/TD split is also useful and leads to a clean empirical result: performance falls sharply once models must read variables or quantitative data from figures. Presentation is mostly good. The paper is easy to follow, the main tables are readable, and the grader comparison appendix is a useful side contribution because it suggests Gemini-2.5-Flash aligns better with human grading than GPT-4o on the tested IPhO cases [1][2][3].

The weaknesses are mostly about how far the claims can go. The medal thresholds are built in incompatible ways across exams: theory-only medalist scores for IPhO/APhO/EuPhO, Hong Kong SAR-only results for PanPhO/PanMechanics, and inferred score lines for F=MA. Because of this, the headline claim of direct human-level comparison is overstated. Soundness is further limited by narrow grader validation (two IPhO exams and two strong closed models), no variance reporting despite eight runs, and no check for possible grader-family bias with Gemini-2.5-Flash grading Gemini-family models. Originality is moderate rather than high: this is solid benchmark construction with one good methodological idea, not a major conceptual advance. The related-work discussion should also mention PhysicsArena and the recent Olympiad-grading framework of McGinness [4], and the paper should state a dataset/code release plan for reproducibility.

[1] UGPhysics: A Comprehensive Benchmark for Undergraduate Physics Reasoning with Large Language Models. Xin Xu et al. ICML 2025. https://arxiv.org/abs/2502.00334

[2] Physics: Benchmarking Foundation Models on University-Level Physics Problem Solving. Kaiyue Feng et al. Findings of ACL 2025. https://aclanthology.org/2025.findings-acl.610/

[3] PhysicsArena: The First Multimodal Physics Reasoning Benchmark Exploring Variable, Process, and Solution Dimensions. Song Dai et al. Findings of EMNLP 2025. https://aclanthology.org/2025.findings-emnlp.937/

[4] Overview of AI Grading of Physics Olympiad Exams. Lachlan McGinness. AIED Doctoral Consortium / arXiv:2505.02121, 2025. https://arxiv.org/abs/2505.02121

---

> ### Author Rebuttal · Authors · 2026-03-31
>
> We appreciate the reviewer's positive feedback on the novelty, evaluation design, and clarity of presentation. We will revise the paper, including a complete related-work discussion, and will open-source the data and code upon acceptance.
>
> ---
>
> **Comment 1.** Did you compare Gemini-family model outputs under Gemini-2.5-Flash and a non-Gemini grader?
>
> **Response 1.** We addressed it in Appendix E.3 (Table 12), where we evaluated four graders: Gemini-2.5-Flash, DeepSeek-R1, GPT-4o, and human experts. The results show that **Gemini-2.5-Flash is closely aligned with human scoring**, DeepSeek-R1 also performs well, while GPT-4o exhibits larger deviations.
>
> We further examined potential grader bias by comparing the **mean ± std over 8 runs**. Because DeepSeek-R1 API was rate-limited, we use DeepSeek-V3.2 instead. As shown in Table R1, DeepSeek-V3.2 and Gemini-2.5-Flash produce broadly similar scores within variance, suggesting that **Gemini-2.5-Flash does not exhibit an obvious bias toward Gemini-family models**.
>
> **Table R1. Mean ± std over 8 runs on IPhO-2025.**
>
> |Grader|Gemini-2.5-Flash|DeepSeek-V3.2|
> -|-:|-:
> Gemini-3-Pro|25.2 ± 0.6|24.1 ± 0.9
> Gemini-2.5-Pro|22.7 ± 0.8|21.2 ± 1.0
> GPT-OSS-120B|20.5 ± 0.7|20.2 ± 1.0
> Qwen3-VL-30B-A3B-Thinking|15.7 ± 2.3|13.0 ± 1.7
>
> ---
>
> **Comment 2.** What are the stds or confidence intervals over the 8 runs for the top models?
>
> **Response 2.** We now report results as **mean ± std over 8 runs** (Table R2), and provide **all 8 runs** (Table R3). We will include these in the revision, showing that the evaluation is statistically reliable.
>
> - **IPhO-2025 differences are clearly beyond noise.** For example, Gemini-3-Pro achieves **25.2±0.6** and Gemini-2.5-Pro **22.7±0.8**, a **2.5-point gap** well above variance. Their 95% CIs do not overlap, and Gemini-3-Pro outperforms Gemini-2.5-Pro in all 8 runs (Table R3), confirming a robust scoring on IPhO-2025.
>
> - **Medal ranking reduces score-level bias.** When performance gaps are small, raw score rankings can be noise-sensitive. Our leaderboard (Table 3) uses **medal counts rather than raw scores**, treating statistically similar models (e.g., all gold) equivalently. The final ranking aggregates **medal counts across exams**, yielding a more stable and robust measure of overall performance than single-exam score ranking.
>
> **Table R2. Mean ± std over 8 runs for top models.**
>
> |Exam|Avg. score / problem|Gemini-3-Pro|GPT-5.2|Gemini-2.5-Pro|GPT-OSS-120B|
> -|:-:|:-:|:-:|:-:|:-:
> IPhO-2025|0.75|25.2 ± 0.6|22.3 ± 0.4|22.7 ± 0.8|20.5 ± 0.7
> NBPhO-2025|2.18|37.1 ± 1.4|35.8 ± 2.3|32.3 ± 1.8|32.4 ± 1.1
>
> **Table R3. All 8 runs on IPhO-2025.**
>
> |Model|Mean|Std|95% CI|Run 1|Run 2|Run 3|Run 4|Run 5|Run 6|Run 7|Run 8|
> -|-:|-:|-:|-:|-:|-:|-:|-:|-:|-:|-:
> Gemini-3-Pro|25.2|0.6|[24.8, 25.6]|24.5|26.5|25.0|25.2|25.5|25.5|24.7|25.1
> Gemini-2.5-Pro|22.7|0.8|[22.1, 23.3]|21.6|21.8|22.9|22.3|24.1|22.9|22.8|23.5
>
> ---
>
> **Comment 3.** How many actual medalists fall below your theory-only cutoff, and is the cutoff driven by an outlier with an unusually strong experimental score?
>
> **Response 3.** We verify that the theory-only cutoff is not driven by an experiment-heavy outlier using both sensitivity analysis and statistical tests.
>
> First, in a leave-one-out analysis (removing the gold medalist with the lowest theoretical score), the cutoff changes only marginally:
> - **IPhO-2025:** original cutoff exceeds **82.7%** of medalists (**90.9%** of contestants); leave-one-out cutoff exceeds **79.0%** of medalists (**88.9%** of contestants); only **2/406 contestants** lie between the two cutoffs.
> - **APhO-2025:** only **3/240 contestants** lie between the two cutoffs.
> - **EuPhO-2025:** only **3/189 contestants** lie between the two cutoffs.
>
> Second, using the IQR rule, the lowest-theory gold medalist in IPhO-2025 is not an outlier: the theory score (19.7) is above the lower bound (16.3), and the experimental score (14.9) is below the upper bound (15.5). The same holds for APhO and EuPhO, indicating that the cutoff is not driven by an outlier.
>
> Given this, our definition of the cutoff is well-justified. We define gold-level theoretical performance as matching or exceeding the theoretical score of a gold medalist, and accordingly set the cutoff to the **minimum theoretical score among gold medalists**. This provides a conservative threshold consistent with Olympiads, where gold medalists are expected to be strong in both theory and experiment.
>
> ---
>
> **Comment 4.** Do the main conclusions still hold on the 7 exams with official marking schemes?
>
> **Response 4.** We compared the leaderboard on the **7 exams with official rubrics** against that on the **6 exams without official rubrics**. The overall medal-tier ranking remains largely unchanged: the top models stay in the same leading group, the middle models remain in the same competitive band, and the lower-performing models also preserve their relative positions. This consistency suggests that our main conclusions are robust.

---

> > ### Author Rebuttal · Reviewer_83rP · 2026-04-08
> >
> > I have carefully considered the authors’ rebuttal and the discussion with the other reviewers. However, I will maintain my original score.

---

### Decision · Program_Chairs · 2026-04-30

**Decision:**

Accept (regular)

**Comment:**

This paper introduces a new reasoning benchmark called HiPhO, for high-school physics Olympiads, which compiles 13 recent Olympiad exams in a multimodal fashion. The benchmark’s uses official rubrics for step-level grading and evaluates 30 state-of-the-art (M)LLMs, reporting medal counts and analyses across modality types, physics fields, and step-level error categories.

Reviewers found that this work covered an important gap in the existing literature, focusing on recent physics Olympiad problems that are multimodal and graded with official rubrics. The use of step-level grading with official marking schemes, which better matches real Olympiad evaluation, appears to provide additional signal. The TO/TI/TV/TD taxonomy is also valuable: models struggle much more when they must extract variables or quantitative information from figures. The authors also presented a strong rebuttal which addressed most stated reviewer concerns.

Overall, the meta-reviewer recommends acceptance of the paper.